# RelaxFlow: Text-Driven Amodal 3D Generation

Jiayin Zhu [1]  Guoji Fu [1]  Xiaolu Liu [2 1]  Qiyuan He [1]  Yicong Li [3]  Angela Yao [1]

## Abstract

Image-to-3D generation faces inherent semantic ambiguity under occlusion, where partial observation alone is often insufficient to determine object category. In this work, we formalize *text-driven amodal 3D generation*, where text prompts steer the completion of unseen regions while strictly preserving input observation. Crucially, we identify that these objectives demand distinct control granularities: rigid control for the observation versus relaxed structural control for the prompt. To this end, we propose **RelaxFlow**, a training-free dual-branch framework that decouples control granularity via a Multi-Prior Consensus Module and a Relaxation Mechanism. Theoretically, we prove that our relaxation is equivalent to applying a low-pass filter on the generative vector field, which suppresses high-frequency instance details to isolate geometric structure that accommodates the observation. To facilitate evaluation, we introduce two diagnostic benchmarks, **ExtremeOcc-3D** and **AmbiSem-3D**. Extensive experiments demonstrate that RelaxFlow successfully steers the generation of unseen regions to match the prompt intent without compromising visual fidelity. Code: https://github.com/viridityzhu/RelaxFlow.

## 1. Introduction

Humans possess the natural ability to perceive objects' complete physical structure even under occlusion. This cognitive capacity, known as amodal perception, allows us to mentally complete the object by supplementing the visible fragments with internal prior knowledge. As a core technology for AR/VR and robotics (Xiu et al., 2025a), image-to-3D generation (Xiang et al., 2025; Zhao et al., 2025; Chen et al.,

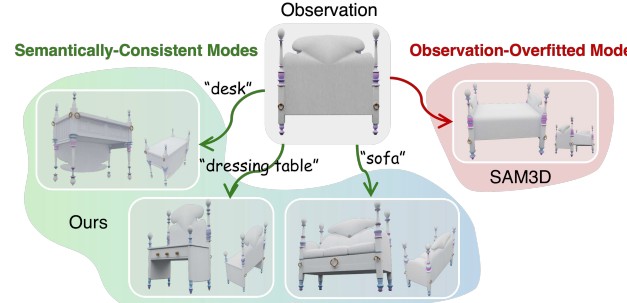

*Figure 1.* The case for multiple plausible amodal 3D interpretations under occlusion. Feedforward image-to-3D model (*e.g.*, SAM3D) collapses to a single overfitted bed-like shape, while our RelaxFlow resolves the ambiguity via *text-driven amodal generation*, allowing users to inject explicit textual intent and steer the generation toward alternative *semantically consistent* amodal 3D shapes.

2025a;b; Liu et al., 2023; Ye et al., 2025; Wu et al., 2024; Poole et al., 2023) seeks to emulate this capability. However, unlike human cognition, which draws on rich internal context, existing generative models typically rely solely on the visible pixels. Consequently, when observation is occluded, this lack of semantic guidance leads to severe ambiguity.

In this work, we move beyond generic single-condition generation and define *text-driven amodal 3D generation*, where the user's text prompt explicitly steers the completion of unseen parts while preserving the observed evidence. Taking Figure 1 as an illustration, when only a wooden backboard is visible, the amodal object remains ambiguous. The unseen regions could plausibly correspond to a sofa, a bed, or a dressing table. Previous feedforward image-to-3D models, like SAM3D, often collapse to an *observation-overfitted* generation (here, a bed-like shape) by uncontrolled implicit hallucination. Under our task formulation, a user can explicitly resolve this ambiguity via a text prompt (*e.g.*, specifying "a sofa"), while keeping the visible part consistent with the input observation.

To achieve this controllability, the generation must satisfy two constraints simultaneously: (1) **observation fidelity**, where the visible regions must strictly adhere to the pixel-level details of the input observation; and (2) **prompt following**, where the unobserved regions must conform to the user's prompt, while maintaining structural coherence with the observation. State-of-the-art feedforward models (Xiang et al., 2025; Chen et al., 2025a; Zhao et al., 2025)

[1]National University of Singapore [2]Zhejiang University [3]University of Science and Technology of China. Correspondence to: Yicong Li <liyicong@ustc.edu.cn>.

*Proceedings of the 43$^{rd}$ International Conference on Machine Learning*, Seoul, South Korea. PMLR 306, 2026. Copyright 2026 by the author(s).

prioritize observation fidelity but often trap the output in an "*observation-overfitted mode*" (see Figure 1, "bed" result) without prompt-following capabilities. Conversely, optimization-based methods (Haque et al., 2023; Kamata et al., 2023; Miao et al., 2025; Zhu et al., 2026a) enforce strong prompt-following, but tend to over-smooth or distort the observation evidence, as gradients from the semantic prior clash with the hard pixel reconstruction constraints (He et al., 2023). As a result, the tension between the observation fidelity and prompt following cannot be resolved by existing strategies.

Existing strategies attempt to enforce both constraints under a uniform control granularity. However, this approach creates an inherent conflict: the *input observation* demands strict adherence to ensure visual fidelity, whereas the *input prompt* serves only as a relaxed structural guide to accommodate the observation. To resolve this tension, our core insight is that the generation process must decouple these control granularities. Accordingly, we adopt a dual-branch strategy: an Observation Branch for strict adherence, and a Semantic-Prior Branch that captures global semantics while tolerating local variations. Finally, we employ a visibility-aware fusion mechanism to ensure the semantic guide steers only the occluded regions, while the observation strictly governs the visible pixels.

While the Observation Branch can be effectively managed via standard rigid feature injection, the challenge arises in designing the relaxation mechanism in the Semantic-Prior Branch. To achieve this, we adopt a training-free strategy with two core designs. We first introduce a Multi-Prior Consensus module that retrieves a set of reference images via the text prompt. Since these references share a semantic category but differ in appearance, their cross-attention naturally amplifies structural consensus while suppressing inconsistent instance-specific textures. We then apply a low-pass relaxation by smoothing cross-attention logits within the generation backbone.

Our theoretical analysis shows that this smoothing is equivalent to low-pass filtering the underlying generative vector field. The filtering suppresses high-frequency instance identities and exposes a coarse semantic corridor that enforces only low-frequency global geometry. As a result, the Semantic-Prior Branch achieves the target geometry (*e.g.*, ensuring the shape of a "sofa") but remains agnostic to high-frequency details, thereby tolerating local variations and accommodating the specific textures of the observation.

To facilitate systematic evaluation, we develop two diagnostic benchmarks for text-driven amodal 3D generation, focusing on whether text can disambiguate unseen structure without sacrificing adherence to the observed evidence. *ExtremeOcc-3D* targets extreme occlusion in natural indoor scenes, where visible evidence is often insufficient to in-fer even the object category, and a text label is required to resolve the completion. *AmbiSem-3D* targets semantic branching: each ambiguous input image admits multiple plausible interpretations, paired with textual prompts that specify distinct intended outcomes under the same visual evidence. Results on these benchmarks show that our method produces high-quality 3D assets that faithfully reflect the user's semantic intent while strictly anchoring the visible evidence, and remain 3D plausible and consistent.

Our contributions are fourfold: (1) We formalize text-driven amodal 3D generation, a new setting that requires resolving occlusion-induced ambiguity via text prompts while strictly preserving the input observation; (2) We propose RelaxFlow, a training-free dual-branch framework that effectively decouples control granularity via consensus-based multi-prior conditioning and visibility-aware fusion; (3) We provide a theoretical proof that our relaxation mechanism on the semantic branch is equivalent to a low-pass filter on the generative vector field, justifying our strategy for extracting structural guidance. (4) We introduce two diagnostic benchmarks, ExtremeOcc-3D and AmbiSem-3D. Extensive experiments demonstrate that our approach significantly outperforms existing strategies in steering unseen geometry without compromising fidelity.

## 2. Related Work

**Occlusion-aware 3D Generation.** Feedforward image-to-3D methods (Xiang et al., 2025; Zhao et al., 2025; Chen et al., 2025b; Lan et al., 2024; Xu et al., 2024) enable efficient single-view generation, but often rely on strong observation priors and degrade under heavy occlusion, where multiple 3D hypotheses remain consistent with the visible evidence. This is closely related to amodal completion, which recovers occluded object extent (Li et al., 2025). In 2D, Ao et al. (2025); Xiu et al. (2025b) present training-free text-guided amodal completion; Pix2Gestalt (Ozguroglu et al., 2024) completes in 2D and lifts to 3D via reconstruction. Occlusion completion has also been integrated into feedforward image-to-3D. Amodal3R (Wu et al., 2025) and DeOcc-1-to-3 (Qu et al., 2025) finetune feedforward models with masked inputs, while SAM3D (Chen et al., 2025a) scales training for stronger completion. However, these approaches require retraining and typically learn a dataset-driven "most likely" completion, limiting controllability when multiple amodal interpretations are plausible. Achieving retraining-free external semantic control (text or images) while preserving observation fidelity remains insufficiently addressed, especially under extreme occlusion.

**Multi-condition Control in 3D Generation.** A common way to reduce ambiguity is to provide more views, *i.e.*, multi-view-to-3D methods (Liu et al., 2023; Shi et al., 2024); however, they assume mutually consistent views of the same

instance, rather than an auxiliary condition that encodes semantic intention. Text-guided 3D editing (Haque et al., 2023; Kamata et al., 2023; Miao et al., 2025; Parelli et al., 2025; Li et al., 2024; 2025) offers language control but typically requires an existing 3D asset and per-instance optimization or inversion, differing from feedforward generation conditioned on an observed image plus an intention signal. More broadly, feedforward image-to-3D generators (Chen et al., 2025a; Zhao et al., 2025; Xiang et al., 2025; Lan et al., 2024) are not designed for unified text-and-image control, making joint alignment of observation evidence and free-form instructions challenging. A practical alternative is to use text-to-image models (Cai et al., 2025; Labs et al., 2025; Podell et al., 2023) to synthesize visual proxies of text intentions that can condition or guide 3D generation.

**Composing and Steering Guidance in Flow Models.** Classifier-free guidance (CFG) (Ho & Salimans, 2022) amplifies conditional signals during diffusion sampling and is often extended to weighted compositions of multiple prompts. Methods that modify condition-dependent dynamics, such as FlowEdit (Kulikov et al., 2025), further suggest that composing conditional vector fields can enforce multiple constraints (Liu et al., 2022). Orthogonally, Hong (2024) analyzes attention blur and uses it as regularizing guidance, and Sadat et al. (2025) studies frequency-aware guidance and sampling. However, a systematic link between attention smoothing and an explicit low-pass semantic prior for stable, training-free controllable generation under severe underdetermination remains underexplored.

## 3. ODE Flow Formulation for Dual-Branch 3D Generation

### 3.1. ODE Flow Formulations

**Motivation: The dual-objective inference problem.** We view text-driven amodal 3D generation as a dual-objective inference problem involving two distinct physical forces:

1. *Observation Constraint:* A hard force that anchors the generation to the visible pixels, ensuring reconstruction fidelity.

2. *Semantic Prior:* A soft force that completes the unobserved regions, governed by the user's high-level semantic intent rather than pixel evidence.

Under external blockers or self-occlusion, the same observation can admit multiple plausible 3D completions, so the semantic completion cannot be uniquely determined from the observation alone.

**Information gap.** The observation condition $c_{\text{obs}}$ (the input image) is sufficient to enforce the *hard* constraint term, but is generally *insufficient* to uniquely determine the semantic

prior in occluded regions. We therefore introduce an auxiliary latent condition $c_{\text{sem}}$ (the user's true intent) that is not a function of $c_{\text{obs}}$ in ambiguous cases.

Motivated by these, we formulate the governing ODE for all methods as an interpolation where the semantic term depends on a variable semantic prior $c^\star$:

$$\frac{\mathrm{d}x_t}{\mathrm{d}t} = (1-\alpha_t)\underbrace{\boldsymbol{v}_{\text{obs}}(x_t, t, c_{\text{obs}})}_{\text{Observation Constraint}} + \alpha_t \underbrace{\boldsymbol{v}_{\text{prior}}(x_t, t, c^\star)}_{\text{Intent Prior (semantic)}}. \quad (1)$$

Here, $\alpha_t$ schedules how strongly the intent prior influences the trajectory. The distinction between methods lies in different choices of (i) the conditioning $c^\star$ used by the prior branch, and (ii) the parameterization of the prior-induced vector field $\boldsymbol{v}$.

**Three instantiations.** *(i) Oracle trajectory.* The ideal flow uses a perfect decomposition: an observation-preserving field and an oracle semantic field driven by the true intent tokens $c^\star = c_{\text{sem}}$. This trajectory has access to the missing information needed to resolve occlusion-induced ambiguity. For the *oracle* semantic transport field, we denote $v_{\text{prior}}(\cdot, t, c^\star) = v_{\text{sem}}(\cdot, t, c_{\text{sem}})$.

*(ii) Standard neural flow.* Feedforward image-to-3D models typically reuse observation tokens for everything, effectively setting $c^\star = c_{\text{obs}}$. This conflates "what must be preserved" with "what is missing," and can yield *observation-overfitted* solutions when the unseen regions are underdetermined. Here, we denote $v_{\text{prior}}(\cdot, t, c^\star) = \bar{v}_\theta(\cdot, t, c_{\text{obs}})$.

*(iii) RelaxFlow (ours).* We keep the observation branch unchanged to preserve evidence, but replace the prior branch with an *explicit intent prior* represented by visual tokens derived from the text prompt, *i.e.*, $c^\star = c_{\text{prior}}$ (Sec. 4.2). Crucially, we *relax* the prior-conditioned estimator with a low-pass operator $\mathcal{R}_\sigma$ (Eq. 2) to suppress instance-specific, high-frequency nuisance that destabilizes joint guidance; in practice $\mathcal{R}_\sigma$ is realized by attention-logit blurring (Eq. 8). This decoupling bridges the information gap without retraining the generator.

Compared to the standard neural flow, RelaxFlow changes only the *source* and *smoothness* of the semantic term: it is driven by intent priors rather than observation tokens, and is low-pass–relaxed to remain compatible with the observation constraint. The next section shows that this relaxation provably reduces semantic estimation error and tightens a stability bound.

### 3.2. Theoretical Justification: Low-Pass Relaxation and Stability

**Low-pass relaxation operator.** Our analysis treats *low-pass relaxation* as an operator applied to the *prior-conditioned* semantic estimator. Let $v_\theta(x, t, c_{\text{prior}})$ denote

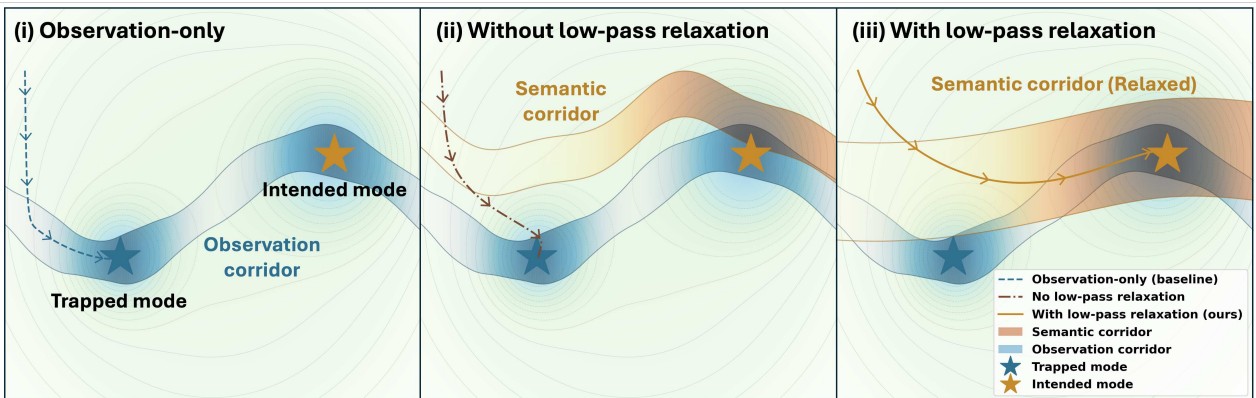

*Figure 2.* **Conceptual illustration of low-pass relaxation.** The background depicts a conceptual *compatibility landscape* over latent states, where high density corresponds to low "energy". Star markers indicate a spurious trapped mode and the intended target mode. Throughout, we use *corridor* to denote a tube of latent states traced by integral curves of a conditioned velocity field while satisfying a constraint. Smoothing the prior-conditioned guidance thickens the semantic corridor and steers trajectories toward the intended mode while remaining compatible with the observation corridor.

the generator's (prior-conditioned) velocity estimate. We define its $\sigma$-relaxed counterpart by

$$\tilde{v}_\theta(\cdot, t, c_{\mathrm{prior}}) := \mathcal{R}_\sigma[v_\theta(\cdot, t, c_{\mathrm{prior}})], \quad (2)$$

where $\mathcal{R}_\sigma$ is a Gaussian low-pass operator that attenuates high-frequency components and preserves low-frequency structure (formalized in Appendix A.2). In the main text, we analyze $\mathcal{R}_\sigma$ abstractly at the velocity-field level; in implementation, we realize $\mathcal{R}_\sigma$ by blurring the *prior-branch* cross-attention logits (Eq. 8), which induces an analogous relaxation on the resulting velocity field (Appendix A.4).

### 3.2.1. SPECTRAL ANALYSIS OF SEMANTIC GUIDANCE

We illustrate the intuition in Figure 2, where the landscape background serves as a conceptual visualization of compatibility. In panel (i), the blue band shows the *observation corridor* induced by $c_{\mathrm{obs}}$, and the dashed curve is an observation-only rollout constrained to remain within this tube. Panels (ii–iii) additionally introduce an intent prior $c_{\mathrm{prior}}$ (orange) to resolve ambiguity in occluded regions: with a raw prior-conditioned estimator (ii), instance-specific high-frequency nuisance can intermittently point against observation-consistent directions, fragmenting the effective semantic corridor and causing unstable dynamics; with low-pass relaxation (iii), these high-frequency components are suppressed, producing a thicker, smoother semantic corridor that stays compatible with the observation corridor and reliably steers the trajectory toward the intended mode.

Motivated by this corridor-fragmentation viewpoint, we model the semantic branch as providing a *transport or steering velocity* in the generator's ODE. We posit that the oracle semantic transport field $v_{\mathrm{sem}}$, which captures global structures like the shape of a "bed" or "sofa", is inherently band-limited to low frequencies. In contrast, errors in the

learned semantic velocity (*e.g.*, texture-driven conflicts or instance-specific hallucinations) behave as high-frequency noise (see Assumptions A.2 and A.3 in Appendix A.2 for formal spectral definitions).

Based on this spectral separation, we treat $\tilde{v}_\theta = \mathcal{R}_\sigma[v_\theta]$ (Eq. 2) as a low-pass relaxation: it suppresses high-frequency, instance-specific error while preserving low-frequency semantic transport. We measure semantic estimation error along a reference trajectory $X_t$ by the $L_2$ path norm:

$$\mathcal{E}_{\mathrm{sem}}^2(v) := \int_0^1 \left\| \boldsymbol{v}_{\mathrm{sem}}(X_t, t, c_{\mathrm{sem}}) - v(X_t, t) \right\|^2 \mathrm{d}t, \quad (3)$$

where $v(X_t, t)$ denotes the semantic-branch estimator used by a method (standard or ours). As derived in Theorem A.4, this operation strictly reduces the semantic estimation error:

$$\mathcal{E}_{\mathrm{sem}}(\tilde{\boldsymbol{v}}_\theta) < \mathcal{E}_{\mathrm{sem}}(\bar{\boldsymbol{v}}_\theta). \quad (4)$$

This denoising operation ensures that the prior branch provides stable structural guidance without injecting the high-frequency conflicts that typically disrupt observation constraints, matching the corridor-smoothing effect in Figure 2. We refer to Appendix A.2 for the detailed derivations.

### 3.2.2. STABILITY AND WASSERSTEIN BOUNDS

We translate the reduction in vector field error into a guarantee on generation quality. We prove that by filtering high-frequency semantic noise, the generated distribution becomes geometrically closer to the ground truth.

We now translate semantic-field estimation error into a distributional guarantee. We use $\mathcal{E}_{\mathrm{sem}}(\cdot)$ as defined in Eq. (3). To measure the quality of the final output, we use the *Wasserstein-2 ($\mathcal{W}_2$) distance* to the ground truth distribution.

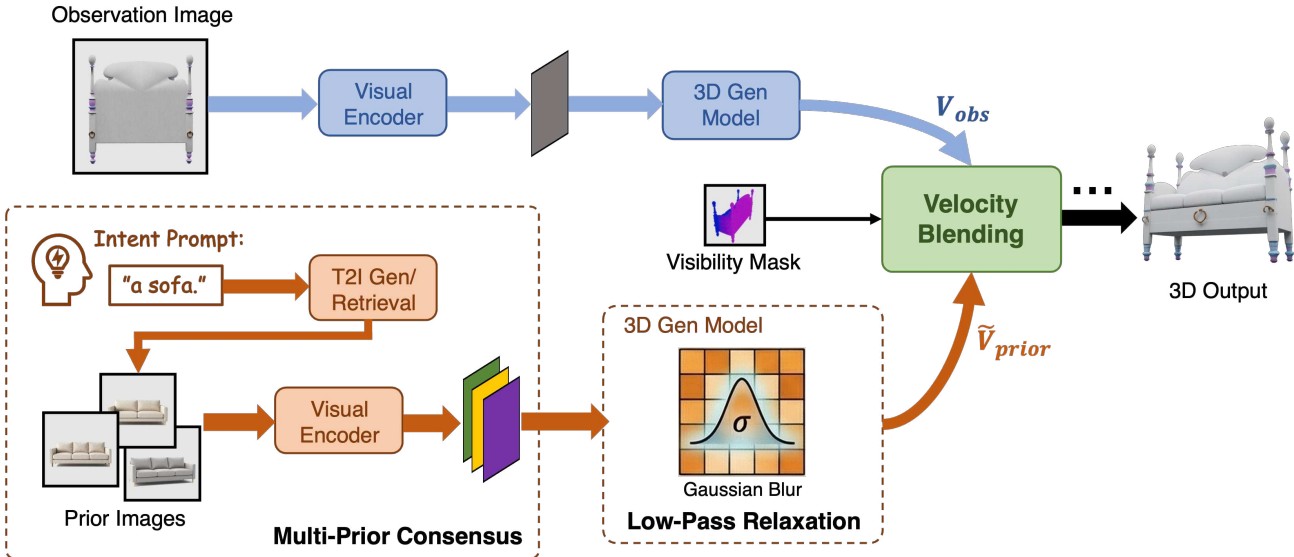

*Figure 3.* **RelaxFlow pipeline overview**. An observation-driven branch preserves visible evidence, while the semantic guidance is injected through the intent prompt via multi-prior consensus and low-pass relaxation. Dual branches are fused via velocity blending to resolve occlusion-induced ambiguity.

We choose $\mathcal{W}_2$ because it captures the *geometric transport cost* between distributions (Heusel et al., 2017a), ensuring that generated shapes are geometrically plausible rather than just statistically overlapping.

Relying on the Lipschitz continuity of the neural estimator (see Assumption A.5 in Appendix), we derive an upper bound for the generation quality. As proven in Theorem A.9 (Appendix), the distance to the ground truth is bounded by

$$\mathcal{W}_2(p, \hat{p}) \leq C \cdot (\mathcal{E}_{\text{obs}}(\bar{\boldsymbol{v}}_\theta) + \mathcal{E}_{\text{sem}}(\tilde{\boldsymbol{v}}_\theta) + \delta_{\text{prior}}). \quad (5)$$

The term $\delta_{\text{prior}}$ captures the residual mismatch between the visual prior tokens $c_{\text{prior}}$ and the oracle intent $c_{\text{sem}}$ (*i.e.*, the gap introduced by using proxy priors). Low-pass relaxation tightens the bound by reducing the semantic estimation error $\mathcal{E}_{\text{sem}}^2(\tilde{\boldsymbol{v}}_\theta)$. In Sec. 4.2, we further reduce $\delta_{\text{prior}}$ in practice by using *multiple* priors and letting attention implement a consensus over consistent attributes.

## 4. RelaxFlow Framework

We introduce RelaxFlow, a training-free dual-branch inference framework for text-driven amodal 3D generation that realizes the theoretical flow derived in Sec. 3.1. As shown in Figure 3, our pipeline decouples the generation into two parallel trajectories: (1) An *Observation Branch* driven by $c_{\text{obs}}$ to preserve high-frequency visible evidence; and (2) A *Semantic-Prior Branch* driven by $c_{\text{prior}}$ that applies low-pass relaxation to steer global structure. We describe the construction of the prior condition $c_{\text{prior}}$ (Sec. 4.2), the realization of the relaxed vector field $\tilde{v}$, and the fusion strategy of the two branches (Sec. 4.3).

### 4.1. Backbone and Inference Setup

We instantiate RelaxFlow on two common feedforward image-to-3D generators, TRELLIS (Xiang et al., 2025) and SAM3D (Chen et al., 2025a), both of which follow a two-stage pipeline: *Sparse Structure* (SS) for generating geometry, and *Structured Latent* (SLAT) for refining texture and detail. The base generators of both SS and SLAT are conditional rectified-flow models. During inference, we solve its ODE with an explicit solver. With $K$ steps and schedule $\{t_k\}$, Euler updates take the form $x_{k+1} = x_k + \Delta t\, v(x_k, t_k, c)$, where $c = E(I, M)$ denotes conditioning tokens encoded from an image-mask pair via a visual encoder $E$, and $v$ is the model-predicted velocity. In SS, a flow generator outputs a latent $z^{\text{ss}}$ decoded into an occupancy volume $S = \mathsf{D}_{\text{ss}}(z^{\text{ss}})$ on a $64^3$ grid, from which we extract occupied voxel indices

$$\mathcal{C} = \{i \mid S_i > 0\}. \quad (6)$$

In SLAT, a second flow generator outputs per-voxel features $z^{\text{slat}} \in \mathbb{R}^{|\mathcal{C}| \times 8}$, and a decoder maps these latents to Gaussian splats or a mesh representation.

### 4.2. Multi-Prior Consensus

A key practical challenge in text-driven amodal 3D generation is that modern feedforward 3D generators are typically *visual-token conditioned* (He et al., 2025): their conditioning interface expects image-derived tokens rather than free-form language. Directly injecting a text embedding would require aligning an external language space to the generator's internal visual tokens (*e.g.*, via adapters or finetuning),

which is both model-specific and risks introducing distribution mismatch. We instead convert the user's intent prompt into *visual proxies* and keep *all* conditions in the generator's native visual-token space.

**Prompt-to-prior as a visual interface.** Given an intent prompt $p$, we construct a small set of $N$ prior images $\{(I_p^n, M_p^n)\}_{n=1}^N$ using either retrieval from a data pool or an off-the-shelf text-to-image generator to sample multiple instances. These priors are not meant to match the observation's instance appearance; rather, they provide *structural semantic evidence* for the occluded or ambiguous regions. The same mechanism also supports user-provided reference images by directly treating them as priors.

**Consensus over multiple priors.** A single prior image can entangle the intended attribute with incidental, instance-specific texture and style details. We therefore condition on a *set* of priors that share the intended attribute but vary in irrelevant factors (*e.g.*, given a prompt "a bird with a red beak", we can generate multiple images with red beaks but different body shapes and feather colors). The model should preserve what is consistent across priors and ignore conflicts. Concretely, we concatenate the per-prior token sequences into one long sequence and feed it to cross-attention in a single pass. This induces a simple consensus effect: attributes that are consistent across priors appear repeatedly in the token set and thus receive more aggregate attention, while idiosyncratic or conflicting details tend to be diluted. Empirically, this reduces the proxy gap $\delta_{\text{prior}}$ in Sec. 3.2 by making the effective conditioning closer to the user's intent than any single prior.

### 4.3. Dual-Branch Sampling as ODE Interpolation

RelaxFlow implements Eq. 1 by coupling two branches without retraining, implemented via a prior-guided sampling rule. At each solver step $k$, we evaluate both branches on the *same* state $x_k$:

$$v_{\text{obs}} = v_\theta(x_k, t_k, c_{\text{obs}}), \quad \tilde{v}_{\text{prior}} = \tilde{v}_\theta(x_k, t_k, c_{\text{prior}}), \quad (7)$$

where $\tilde{v}_\theta$ denotes evaluation under low-pass relaxation, which we implement by smoothing the cross-attention logits as in Eq. 8. This shared-state design matters: it avoids inconsistent trajectories (*e.g.*, a prior-only path that drifts away from the observed geometry), and instead combines two compatible velocities.

**Realizing $\tilde{v}$ via logit smoothing.** To instantiate the abstract relaxation $\tilde{v}_\theta = \mathcal{R}_\sigma[v_\theta]$ (Eq. 2) in transformer-based generators, we apply $\mathcal{R}_\sigma$ to the *prior-branch* cross-attention *logits*. This blurring suppresses token-local, high-frequency sensitivity in prior conditioning, yielding the smoother semantic corridor in Figure 2. For an attention head with

logits $L_{i,j} = q_i^\top k_j / \sqrt{d}$, where $(i, j)$ follow the underlying 2D/3D token grid order, we apply a separable Gaussian filter $G_\sigma$ before the softmax:

$$\tilde{L} = G_\sigma * L := \left( G_\sigma^{(q)} *_q L \right) *_k G_\sigma^{(k)}, \quad (8)$$

and $\text{Attn}(Q, K, V) = \text{softmax}(\tilde{L}) V$. Here $*_q$ and $*_k$ denote 1D convolution along the query and key indices, respectively. In implementation, $G_\sigma$ is a discrete 1D Gaussian kernel with $\pm 3\sigma$ support, normalized to sum to 1; the operation is realized as two sequential 1D convolutions along the sequence dimensions ($Q$ and $K$), equivalent to a separable 2D blur over the logit matrix (Hong, 2024). The hyperparameter $\sigma$ controls the smoothing strength, and we empirically show that performance is robust to $\sigma$ around our default choice ($\sigma = 1.0$) (Sec. 5.5). Observation Branch attention is left unmodified.

**Fusion Strategy.** To combine the branches, we employ a time-dependent and spatially-aware fusion scheme.

Prior works (Huang et al., 2024; Choi et al., 2022; Zhu et al., 2026b) show that early steps determine the global semantic mode under ambiguity, so prior guidance is most useful there; later steps are where the model refines geometry and texture details, where we prefer to rely on the observation branch to avoid overpainting and to preserve evidence-consistent high frequencies. As a result, the prior steers where needed (coarse, ambiguous completion) and yields to the observation for what must be preserved (visible structure and fine details). We then discretize the time-dependent gate $\alpha_t$ as $\alpha_k$ and apply Euler updates:

$$x_{k+1} = x_k + \Delta t \left( (1 - \alpha_k) v_{\text{obs}} + \alpha_k \tilde{v}_{\text{prior}} \right), \quad (9)$$

where $\alpha_k = g(k, K)$ is discretized and clamped to $[0, 1]$, and we use a simple linear cutoff schedule:

$$\alpha_k = \begin{cases} 1 - k/K, & k \le \lfloor \rho K \rfloor, \\ 0, & \text{otherwise,} \end{cases} \quad (10)$$

where $\rho$ is the cutoff threshold. Intuitively, this schedule allows the prior $c_{\text{prior}}$ to steer the global semantic mode early, then yield to the Observation Branch with only observation evidence $c_{\text{obs}}$ to refine fine-grained details.

In the SLAT stage, injecting the Semantic-Prior Branch uniformly risks overpainting regions that are directly supported by the observation. We therefore estimate per-voxel visibility from the observation camera and use it to spatially separate the conditions: observed surfaces are preserved while the prior steers only genuinely occluded regions. Given the object pose from the backbone, we project voxel centers to the image, build a z-buffer depth map $D'$, and compute an occlusion margin $\Delta_i = z_i - D'(u_i, v_i)$ for each voxel $i$ projected to pixel $(u_i, v_i)$ with depth $z_i$. This margin

is converted into a soft visibility weight $m_i \in (0, 1]$ via a Gaussian falloff (details in Appendix F). We apply the visibility mask to the *velocity interpolation* in SLAT:

$$v_i = v_{\text{obs},i} + (1 - m_i)\,\alpha_k\,(\tilde{v}_{\text{prior},i} - v_{\text{obs},i}), \qquad (11)$$

and update $x_{k+1} = x_k + \Delta t\, v$ as in Eq. 9. If $m_i \approx 1$ (visible), then $v_i \approx v_{\text{obs},i}$; if $m_i \approx 0$ (occluded), the step follows the time-varying blend controlled by $\alpha_k$.

# 5. Experiments

We evaluate RelaxFlow on two complementary tasks: (i) amodal 3D completion under extreme occlusion (ExtremeOcc-3D), and (ii) text-driven amodal 3D generation under semantic branching (AmbiSem-3D). Both benchmarks are curated for controlled evaluation; full details about dataset curation are in Appendix B.

## 5.1. Task Setting

**ExtremeOcc-3D.** We build ExtremeOcc-3D from 3D-FUTURE/3D-FRONT indoor scenes (Fu et al., 2021b;a), where occlusions arise naturally from realistic furniture arrangements and camera viewpoints. Using ground-truth meshes and poses, we compute an *occlusion ratio* $r$ for each object. When $r \geq 80\%$, the visible evidence is often insufficient even for humans to identify the object type. We filter 264 test cases at this threshold, each paired with a category-level text prior. The task is to generate a complete 3D asset faithful to the visible evidence while conforming to the specified category.

**AmbiSem-3D.** AmbiSem-3D tests whether text can resolve *semantic uncertainty* rather than merely refine appearance. We curate 21 ambiguous cases from ObjaverseXL (Deitke et al., 2023) exhibiting three ambiguity types: masked semantic completion, view-induced identity ambiguity, and intrinsic shape ambiguity. Each case has a single input image and multiple text branches, where each branch corresponds to a distinct but plausible semantic completion. We note that AmbiSem-3D is intentionally multi-solution and lacks a unique 3D ground truth.

**Metrics.** We evaluate observation preservation (min-LPIPS), semantic alignment (CLIP-Score (Radford et al., 2021)), and multi-view realism (FID (Heusel et al., 2017b)) in 2D. In 3D, as strict geometry errors are ill-suited for our task, where multiple completions are all considered valid, we instead evaluate set-level *semantic* similarity via Point-FID (Nichol et al., 2022), which leverages the Point-E encoder to extract semantic features of point clouds. Details of metric computation are provided in Appendix D.

*Table 1.* **Quantitative comparison on ExtremeOcc-3D.** Methods are grouped by occlusion awareness. Our method improves over both backbones across all metrics.

| Category | Method | CLIP$_{\text{img}}\uparrow$ | CLIP$_{\text{txt}}\uparrow$ | FID$\downarrow$ | LPIPS$\downarrow$ | Point-FID$\downarrow$ |
|---|---|---|---|---|---|---|
| Non-Occlusion Aware | TRELLIS | 0.78 | 23.14 | 122.68 | 0.83 | 141.48 |
| | **Ours (w/ TRELLIS)** | **0.80** | **24.09** | **100.75** | **0.80** | **97.79** |
| Occlusion-Aware | Amodal2D+SAM3D | 0.76 | 21.59 | 94.38 | 0.56 | 127.27 |
| | Amodal3R | 0.77 | 22.29 | 118.49 | 0.60 | 129.46 |
| | SAM3D | 0.84 | 24.08 | 50.73 | 0.54 | 100.38 |
| | **Ours (w/ SAM3D)** | **0.87** | **27.26** | **39.44** | **0.51** | **81.11** |

## 5.2. Implementation Details

We evaluate RelaxFlow as a plug-and-play module on SAM3D (Chen et al., 2025a) and TRELLIS (Xiang et al., 2025). For TRELLIS (no pose estimation), we disable the visibility-aware mask and apply other components unchanged. On ExtremeOcc-3D, we apply RelaxFlow only at the geometry stage; on AmbiSem-3D, we apply it to both stages since semantics affect both structure and appearance. We use $\sigma=1.0$, $\rho=0.2$, and $N=3$ prior images. All experiments run on a single NVIDIA A40 GPU.

**Prior image sources.** For ExtremeOcc-3D, we retrieve prior images by randomly sampling a different object of the same category from the dataset. For AmbiSem-3D, we generate prior images from the text prompt using Z-Image (Cai et al., 2025); all compared methods receive the same generated priors for fair comparison. Details on prior construction are provided in Appendix C.

## 5.3. Amodal 3D Completion Under Extreme Occlusion

**Baselines.** We compare against: (1) non-occlusion-aware methods (TRELLIS (Xiang et al., 2025)), (2) 2D completion pipelines (Amodal2D (Ao et al., 2025)+SAM3D), and (3) occlusion-aware methods (SAM3D (Chen et al., 2025a), Amodal3R (Wu et al., 2025)).

**Results.** Table 1 shows that RelaxFlow delivers consistent gains over its backbone models. On TRELLIS, Point-FID decreases from 141.48 to 97.79. On SAM3D, RelaxFlow achieves the best overall performance: CLIP$_{\text{txt}}$ increases from 24.08 to 27.26, and Point-FID decreases from 100.38 to 81.11. Notably, CLIP$_{\text{img}}$ and LPIPS remain competitive with SAM3D, indicating that RelaxFlow preserves the observed evidence in the input view rather than drifting from it. Meanwhile, the lower FID and Point-FID further suggest improved global consistency across views and higher overall 3D quality. Figure 4 provides qualitative comparisons: TRELLIS exhibits noticeable drift from the observation, SAM3D responds weakly to the textual prompt, and 2D-based editing pipelines often introduce geometric artifacts. In contrast, RelaxFlow preserves the observed structure while following the intended semantics. Overall,

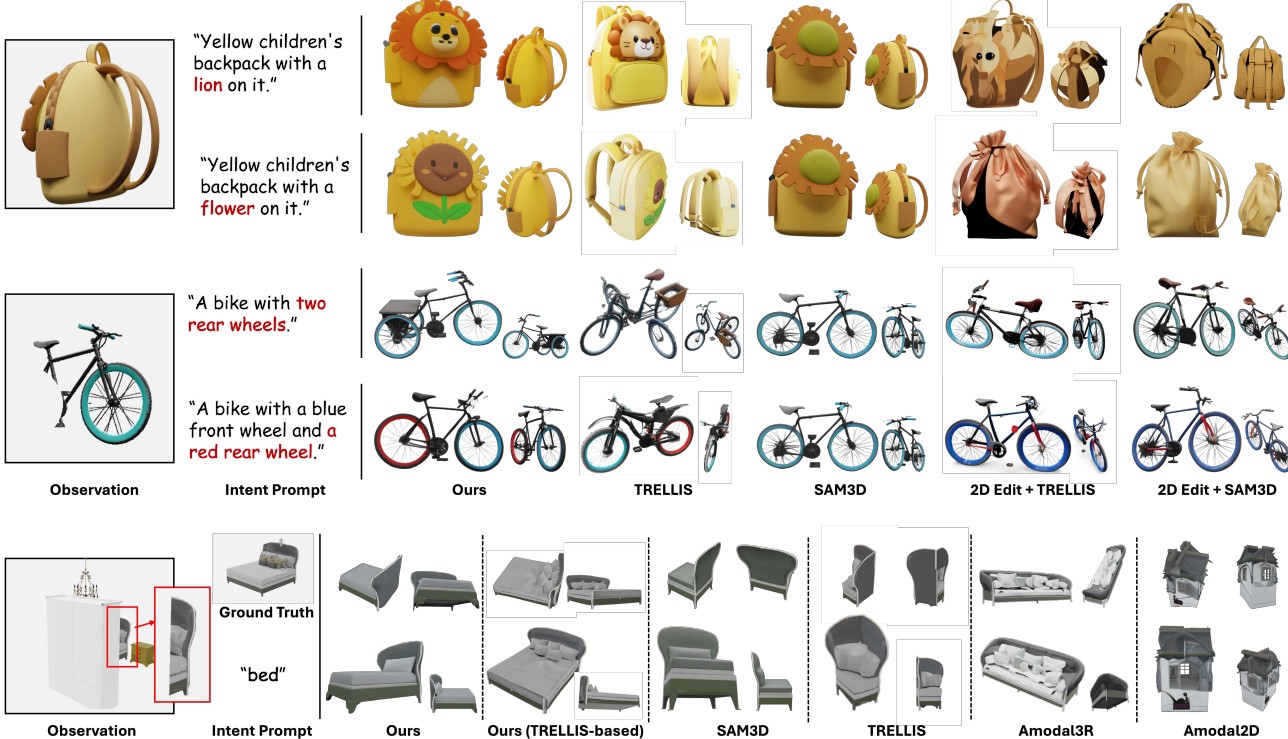

*Figure 4.* **Qualitative Comparisons. Top:** Comparisons on AmbiSem-3D examples. Each contains two different intent prompts for the same observation. Our method preserves observation fidelity while enabling prompt-controlled completion. **Bottom:** The case in ExtremeOcc-3D with an intent prompt "bed". Under extreme occlusion, baselines either overfit the visible region or yield implausible shapes, whereas ours follows the category prior while maintaining visible evidence.

these results indicate that RelaxFlow mitigates observation-overfitted collapse under extreme occlusion.

### 5.4. Text-driven Amodal 3D Generation Under Semantic Branching

**Baselines.** We compare against SAM3D (Chen et al., 2025a), TRELLIS (Xiang et al., 2025), and 2D-edit-then-3D pipelines, where SDXL (Podell et al., 2023) editing is followed by TRELLIS or SAM3D reconstruction. Since TRELLIS accepts multi-view images, we provide the prior image as a second view. Appendix E further compares with commercial 2D editing, video generation, multi-view generation, and direct text+image 3D pipelines.

**Results.** Table 2 and Figure 4 show that RelaxFlow achieves the best alignment to both observation ($\text{CLIP}_{\text{img}}$) and intent prompt ($\text{CLIP}_{\text{txt}}$). Multi-view TRELLIS often drifts from the observation due to conflicting views, while 2D-editing pipelines introduce geometry-inconsistent artifacts.

**User Study.** Since AmbiSem-3D allows intent prompt-driven generations which do not have unique ground truths, we conduct a user study with 32 volunteers evaluating all 21 cases on *Text–Image Alignment* and *3D Fidelity*. As shown in Table 2 (right), RelaxFlow is preferred by a large margin

*Table 2.* **Quantitative comparison on AmbiSem-3D.** Left: automatic metrics (CLIP similarities). Right: user study results ($n{=}32$). Our method achieves the best scores on both evaluations.

| Method | CLIP Score | | User Study | | |
|---|---|---|---|---|---|
| | $\text{CLIP}_{\text{img}}\uparrow$ | $\text{CLIP}_{\text{txt}}\uparrow$ | Alignment$\uparrow$ | 3D Fidelity$\uparrow$ | Overall Pref.$\uparrow$ |
| SAM3D | 0.85 | 26.29 | 4.84% | 13.59% | 9.22% |
| TRELLIS (multi-view) | 0.80 | 26.59 | 3.75% | 8.28% | 6.02% |
| SDXL + TRELLIS | 0.81 | 26.76 | 6.09% | 8.91% | 7.50% |
| SDXL + SAM3D | 0.79 | 26.71 | 11.41% | 6.09% | 8.75% |
| **Ours** | **0.87** | **27.23** | **73.91%** | **63.13%** | **68.52%** |

(68.52% overall), confirming its ability to resolve ambiguity while preserving observed evidence.

### 5.5. Ablation Study

Table 6a and Figure 5 report ablations on ExtremeOcc-3D (SAM3D backbone). Removing low-pass relaxation degrades Point-FID ($81.1{\rightarrow}87.1$), confirming its role in stabilizing semantic guidance. Disabling the visibility mask causes a larger drop ($81.1{\rightarrow}92.3$), showing that spatially isolating occluded regions is critical. Hyperparameter sensitivity: aggressive cutoff ($\rho$) or strong relaxation ($\sigma$) hurts

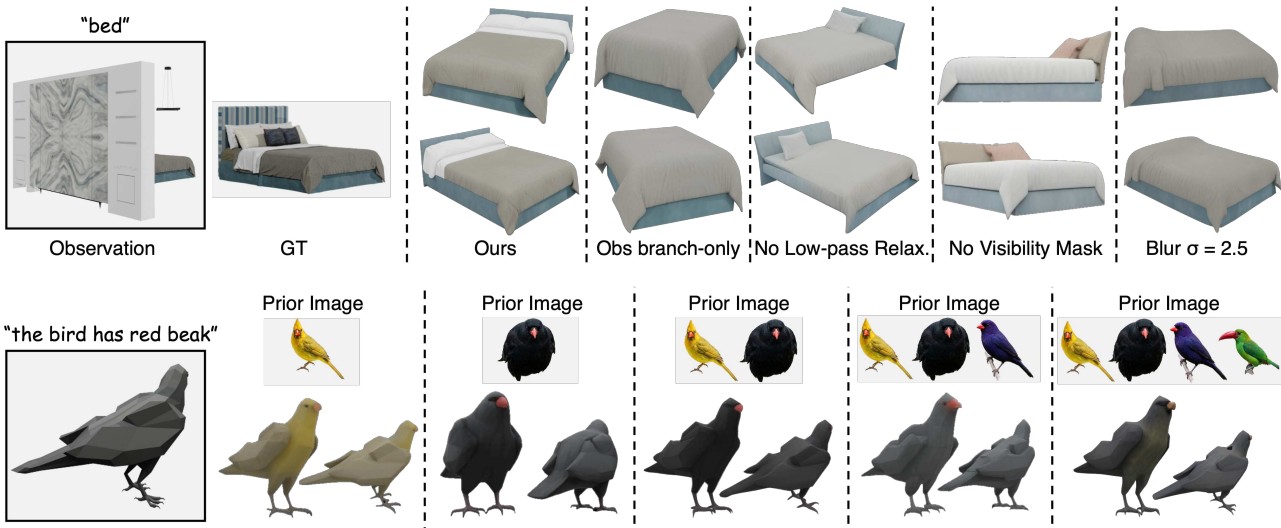

*Figure 5.* **Visual ablation results.** Top: effect of key components. Bottom: effect of varying the number of prior images $N$.

| Setting | Point-FID↓ |
|---|---|
| w/o LP Relax. | 87.1 |
| w/o Visib. Mask | 92.3 |
| cutoff $\rho = 0.4$ | 86.5 |
| cutoff $\rho = 1.0$ | 89.9 |
| LP Relax. $\sigma = 2.5$ | 95.2 |
| Prior from generation | 82.7 |
| **Ours** | **81.1** |

*(a)*

*(b)*

*Figure 6.* **Ablation studies** on ExtremeOcc-3D with SAM3D backbone. (a) Component ablation and hyperparameter sensitivity. Removing low-pass relaxation or visibility mask degrades performance; extreme $\rho$ or $\sigma$ values also hurt. "Prior from generation" uses Z-Image-generated priors instead of retrieved ones. (b) Effect of prior count $N$: moderate $N$ improves consensus, but too many priors introduce conflicts.

performance, balancing semantic signal preservation with high-frequency suppression.

**Number of priors.** Figure 6b shows that moderate $N$ improves consensus-based disambiguation, but excessive priors introduce conflicting details. Generated priors remain competitive (Table in Figure 6a).

**Efficiency.** Appendix E reports the runtime and memory overhead. In brief, the extra branch evaluations increase runtime, while peak memory grows modestly; the lightweight low-pass relaxation itself is implemented with efficient 1D convolutions.

## 6. Conclusion

We introduce text-driven amodal 3D generation, where the same visible evidence admits multiple amodal 3D com-

pletions and users are allowed to specify intent with text prompts. To tackle this, we propose RelaxFlow, a training-free dual-branch inference framework that decouples evidence preservation from semantic disambiguation: the Observation Branch anchors observation-driven high-frequency detail, while we theoretically justify the use of low-pass relaxation in the Semantic-Prior Branch to distill stable, low-frequency semantic guidance from intent-conditioned priors. RelaxFlow further strengthens controllability via multi-prior consensus, and a time-varying visibility-aware fusion schedule. To facilitate systematic evaluation, we introduce ExtremeOcc-3D and AmbiSem-3D, two diagnostic benchmarks that expose observation-overfitted collapse and evaluate intent-following under fixed evidence. Across state-of-the-art feedforward 3D generators, RelaxFlow improves semantic plausibility and controllability without retraining, while preserving observation fidelity.

## Acknowledgements

We would like to acknowledge that computational work involved in this research is partially supported by NUS IT's Research Computing group using grant number NUSREC-HPC-00001. We thank the reviewers and the area chair for their constructive feedback.

## Impact Statement

This work contributes to progress in controllable 3D generation by improving how models handle ambiguity and missing visual evidence. Such advances may support research and applications that rely on reliable 3D content, while lowering barriers to experimentation through training-free methods. At the same time, like other generative tech-

nologies, it could be misused to create misleading synthetic content or amplify biases present in underlying data and models. We encourage responsible use and deployment consistent with applicable policies and norms.

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

# A. Theoretical Analysis of Low-Pass Relaxation via Attention-Logit Smoothing

## A.1. Definitions and Preliminaries

**Definition A.1** (Wasserstein-2 distance). For $\mu, \nu \in \mathcal{P}(\Omega)$, their Wasserstein $k$-distance is defined by

$$W_2(\mu, \nu) = \inf_{\gamma \in \Gamma(\mu, \nu)} \left( \int_{(x,y) \sim \gamma} \|x - y\|^2 \mathrm{d}\gamma(x, y) \right)^{1/2},$$

where $\Gamma(\mu, \nu)$ is the set of all couplings of $\mu$ and $\nu$.

## A.2. Spectral analysis of semantic guidance

We posit that global semantic intent is inherently low-frequency, whereas the errors causing artifacts are high-frequency. This distinction allows us to use filtering to selectively remove errors without degrading the signal. To justify the benefit of Gaussian blur, we characterize the spectral properties of the semantic signal versus the error.

**Assumption A.2** (Spectral concentration of semantic intent). The ideal semantic transport field $\boldsymbol{v}_{\mathrm{sem}}(\cdot, t, c_{\mathrm{sem}})$ is band-limited. In the Fourier domain, its energy is concentrated in low frequencies:

$$\boldsymbol{v}_{\mathrm{sem}}(\omega, t, c_{\mathrm{sem}}) \approx 0 \quad \text{for } |\omega| \geq \eta, \tag{12}$$

where $\eta > 0$ is a frequency cutoff.

Assumption A.2 implies $\boldsymbol{v}_{\mathrm{sem}}$ captures global structure (low-frequency) rather than high-frequency instance details.

For notational convenience, we write

$$\epsilon(x) := \boldsymbol{v}_{\mathrm{sem}}(x) - \bar{\boldsymbol{v}}_\theta(x), \tag{13}$$

$$\mathcal{E}_{\mathrm{obs}}^2(\boldsymbol{v}_\theta) := \int_0^t \left\| \boldsymbol{v}_{\mathrm{obs}}(X_\tau, \tau, c_{\mathrm{obs}}) - \boldsymbol{v}_\theta(X_\tau, \tau, c_{\mathrm{obs}}) \right\|^2 \mathrm{d}\tau, \tag{14}$$

$$\mathcal{E}_{\mathrm{sem}}^2(\boldsymbol{v}_\theta) := \int_0^t \left\| \boldsymbol{v}_{\mathrm{sem}}(X_\tau, \tau, c_{\mathrm{sem}}) - \boldsymbol{v}_\theta(X_\tau, \tau, c_{\mathrm{obs}}) \right\|^2 \mathrm{d}\tau. \tag{15}$$

**Assumption A.3** (High-frequency mismatch). The error between the learned velocity and the ideal semantic velocity is dominated by high-frequency components. Let $\epsilon'$ be the Fourier transform of $\epsilon$. We assume:

$$\int_{|\omega| \leq \eta} |\epsilon'(\omega)|^2 \mathrm{d}\omega \ll \int_{|\omega| > \eta} |\epsilon'(\omega)|^2 \mathrm{d}\omega. \tag{16}$$

**Proposition A.4** (Error reduction via low-pass filtering). *Let $\tilde{\boldsymbol{v}}_\theta = G_\sigma * \bar{\boldsymbol{v}}_\theta$ be the velocity field smoothed by a normalized Gaussian kernel $G_\sigma$. Under Assumptions A.2 and A.3, for a sufficiently small $\beta$, the semantic estimation error of the blurred field is strictly lower than that of the original field:*

$$\mathcal{E}_{\mathrm{sem}}(\tilde{\boldsymbol{v}}_\theta) < \mathcal{E}_{\mathrm{sem}}(\bar{\boldsymbol{v}}_\theta). \tag{17}$$

*Proof.* Notice that

$$\mathcal{E}_{\mathrm{sem}}^2(\bar{\boldsymbol{v}}_\theta) = \int_0^t \|\epsilon(x)\|^2 \mathrm{d}\tau.$$

Let $G_\sigma'(\omega) = \exp(-\frac{1}{2}\sigma^2 \|\omega\|^2) \in (0, 1]$, $\boldsymbol{v}_{\mathrm{sem}}'$, and $\boldsymbol{v}_\theta'$ be the Fourier transform of the Gaussian kernel, $\boldsymbol{v}_{\mathrm{sem}}$, and $\boldsymbol{v}_\theta$, respectively. By Parseval's theorem, the squared $L^2$ error of the smoothed field is:

$$\|\boldsymbol{v}_{\mathrm{sem}} - \tilde{\boldsymbol{v}}_\theta\|_{L^2}^2 = \|\boldsymbol{v}_{\mathrm{sem}}' - G_\sigma' \bar{\boldsymbol{v}}_\theta'\|_{L^2}^2. \tag{18}$$

Using the fact that $\boldsymbol{v}_{\mathrm{sem}}$ is band-limited (Assumption A.2) and that convolution with a normalized Gaussian preserves the low-frequency signal (assuming $G_\sigma'(\omega) \approx 1$ for $\|\omega\| \leq \eta$), we focus on the error term $\epsilon$. The smoothing acts on the error as $\tilde{\epsilon} = G_\sigma * \epsilon$.

We decompose the error energy into low-frequency ($\mathcal{D}_{\text{low}} = \{\omega : \|\omega\| \leq \eta\}$) and high-frequency ($\mathcal{D}_{\text{high}} = \{\omega : \|\omega\| > \eta\}$) regions:

$$\|\tilde{\epsilon}\|_{L^2}^2 = \int_{\mathcal{D}_{low}} |G'_\sigma|^2 \|\epsilon'\|^2 \mathrm{d}\omega + \int_{\mathcal{D}_{high}} |G'_\sigma|^2 \|\epsilon'\|^2 \mathrm{d}\omega$$

$$\leq \int_{\mathcal{D}_{low}} \|\epsilon'\|^2 \mathrm{d}\omega + \gamma_\sigma \int_{\mathcal{D}_{high}} \|\epsilon'\|^2 \mathrm{d}\omega,$$

where $\gamma_\sigma := \sup_{\|\omega\|>\eta} |G'_\sigma(\omega)|^2 < 1$ is the maximum gain in the high-frequency band.

Comparing this to the original error $\|\epsilon\|_{L^2}^2 = \int_{\mathcal{D}_{low}} \|\epsilon'\|^2 + \int_{\mathcal{D}_{high}} \|\epsilon'\|^2$, the reduction is determined by the high-frequency term. Given Assumption A.3, the total error is dominated by the integral over $\mathcal{D}_{high}$. Since $\gamma_\sigma < 1$, the dampening of the dominant high-frequency error strictly outweighs any potential signal loss in the low frequencies (which is negligible or zero under strict band-limiting). Thus, $\|\tilde{\epsilon}\|_{L^2}^2 < \|\epsilon\|_{L^2}^2$, which implies that $\mathcal{E}_{\text{sem}}^2(\tilde{\boldsymbol{v}}_\theta) < \mathcal{E}_{\text{sem}}^2(\bar{\boldsymbol{v}}_\theta)$. $\qquad\square$

*Remark:* In the Fourier domain, the Gaussian kernel $G'_\sigma(\omega) = e^{-\sigma^2\omega^2/2}$ acts as a low-pass filter. Since the semantic signal is low-frequency (Assumption A.2) and the error is high-frequency (Assumption A.3), filtering reduces the error term significantly while preserving the signal.

### A.3. Stability and Wasserstein bounds

In this section, we translate the error reduction derived above into a guarantee on generation quality. We prove that by reducing high-frequency semantic noise, our method produces a distribution that is mathematically closer to the ground truth. We define the accumulated semantic error along the trajectory as $\mathcal{E}_{\text{sem}}^2(\boldsymbol{v}) = \int_0^1 \|\boldsymbol{v}_{\text{sem}}(X_\tau) - \boldsymbol{v}(X_\tau)\|^2 \mathrm{d}\tau$. We rely on the Lipschitz continuity of the neural estimator (Assumption A.5 in Appendix).

**Assumption A.5** (Lipschitz neural network estimator). The neural network estimator $\bar{\boldsymbol{v}}_\theta(x, t, c)$ is $L_t^{\text{nn}}$-Lipschitz with respect to state $x$ and condition $c$:

$$\|\bar{\boldsymbol{v}}_\theta(x, t, c) - \bar{\boldsymbol{v}}_\theta(x', t, c)\| \leq L_t^{\text{nn}}\|x - x'\|,$$
$$\|\bar{\boldsymbol{v}}_\theta(x, t, c) - \bar{\boldsymbol{v}}_\theta(x, t, c')\| \leq L_t^{\text{nn}}\|c - c'\|.$$

**Proposition A.6** (Stability of Lipschitz Constant under Gaussian Smoothing). *Let $v$ be an $L$-Lipschitz function, and let $G_\sigma$ be a normalized Gaussian kernel (i.e., $\int G_\sigma(y)\mathrm{d}y = 1$). The convolved function $\tilde{v} := G_\sigma * v$ is also Lipschitz, satisfying:*

$$\|\tilde{v}\|_{\text{Lip}} \leq \|v\|_{\text{Lip}} = L.$$

*Proof.* For any $x, x'$, we have:

$$\|\tilde{v}(x) - \tilde{v}(x')\| = \left\| \int G_\sigma(y)(v(x - y) - v(x' - y))\mathrm{d}y \right\|$$

$$\leq \int G_\sigma(y)\|v(x - y) - v(x' - y)\|\mathrm{d}y$$

$$\leq \int G_\sigma(y)L\|(x - y) - (x' - y)\|\mathrm{d}y$$

$$= L\|x - x'\| \underbrace{\int G_\sigma(y)\mathrm{d}y}_{=1} = L\|x - x'\|.$$

Thus, $\|\tilde{v}\|_{\text{Lip}} \leq L$. $\qquad\square$

A.3.1. STABILITY ANALYSIS

Consider the ODE flow:

$$\frac{\mathrm{d}x_t}{\mathrm{d}t} = (1 - \alpha_t)\boldsymbol{v}_{\mathrm{obs}}(x_t, t, c_{\mathrm{obs}}) + \alpha_t \boldsymbol{v}_{\mathrm{sem}}(x_t, t, c_{\mathrm{sem}}), \quad x_0 \sim \rho \equiv \mathcal{N}(0, \boldsymbol{I}_d), \tag{19}$$

$$\frac{\mathrm{d}\bar{x}_t}{\mathrm{d}t} = (1 - \alpha_t)\bar{\boldsymbol{v}}_\theta(\bar{x}_t, t, c_{\mathrm{obs}}) + \alpha_t \bar{\boldsymbol{v}}_\theta(\bar{x}_t, t, c_{\mathrm{obs}}), \quad \bar{x}_0 \sim \rho \equiv \mathcal{N}(0, \boldsymbol{I}_d), \tag{20}$$

$$\frac{\mathrm{d}\hat{x}_t}{\mathrm{d}t} = (1 - \alpha_t)\bar{\boldsymbol{v}}_\theta(\hat{x}_t, t, c_{\mathrm{obs}}) + \alpha_t \tilde{\boldsymbol{v}}_\theta(\hat{x}_t, t, c_{\mathrm{prior}}), \quad x_0 \sim \rho \equiv \mathcal{N}(0, \boldsymbol{I}_d). \tag{21}$$

We introduce the following definitions from optimal transport (Peyré & Cuturi, 2018), which are used in the proofs for our theoretical analysis:

- **Integrated ODE flow:** For a velocity field $v$, the state at time $t$ is:

$$X_t = X_0 + \int_0^t v(X_\tau, \tau)\mathrm{d}\tau. \tag{22}$$

- **Measure transportation:**

$$T(x) := x + \int_0^T \frac{\mathrm{d}}{\mathrm{d}\tau} X_\tau \mathrm{d}\tau. \tag{23}$$

- **Pushforward measure:**

$$p_0 = T_\sharp \rho, \tag{24}$$

  where $p_0$ is the underlying data distribution and $\rho$ is the reference distribution, which is generally the standard Gaussian distribution.

- **Pullback measure:**

$$\rho = T^\sharp p_0 = (T^{-1})_\sharp p_0.$$

**Lemma A.7** (Stability analysis for ODE flows). *Let $X_t, \overline{X}_t, \widehat{X}_t$ be the integrals Eq.* (22) *with the ODE flows Eq.* (19) *to* (21)*, respectively. Then we have*

$$\|X_t - \overline{X}_t\| \leq \left((1 - \alpha_t)\mathcal{E}_{\mathrm{obs}}(\bar{\boldsymbol{v}}_\theta) + \alpha_t \mathcal{E}_{\mathrm{sem}}(\bar{\boldsymbol{v}}_\theta) + \|c_{\mathrm{sem}} - c_{\mathrm{obs}}\| \int_0^t L_\tau^{\mathrm{nn}}\mathrm{d}\tau\right)\exp\left(\int_0^t L_\tau^{\mathrm{nn}}\mathrm{d}\tau\right),$$

$$\|X_t - \widehat{X}_t\| \leq \left((1 - \alpha_t)\mathcal{E}_{\mathrm{obs}}(\bar{\boldsymbol{v}}_\theta) + \alpha_t \mathcal{E}_{\mathrm{sem}}(\tilde{\boldsymbol{v}}_\theta) + \|c_{\mathrm{sem}} - c_{\mathrm{prior}}\| \int_0^t \widetilde{L}_\tau^{\mathrm{nn}}\mathrm{d}\tau\right)\exp\left(\int_0^t \left((1 - \alpha_t)L_\tau^{\mathrm{nn}} + \alpha_t \widetilde{L}_\tau^{\mathrm{nn}}\right)\mathrm{d}\tau\right).$$

*Proof.* By the Lipschitz assumptions for $\boldsymbol{v}_{\mathrm{obs}}$ and $\bar{\boldsymbol{v}}_\theta$, we have

$$\|\boldsymbol{v}_{\mathrm{obs}}(X_t, t, c_{\mathrm{obs}}) - \boldsymbol{v}_{\mathrm{obs}}(\overline{X}_t, t, c_{\mathrm{obs}})\| \leq L_t^{\mathrm{obs}}\|X_t - \overline{X}_t\|, \tag{25}$$

$$\|\bar{\boldsymbol{v}}_\theta(X_t, t, c_{\mathrm{sem}}) - \bar{\boldsymbol{v}}_\theta(\overline{X}_t, t, c_{\mathrm{sem}})\| \leq L_t^{\mathrm{nn}}\|X_t - \overline{X}_t\|, \tag{26}$$

**(i) bounding $\|X_t - \overline{X}_t\|$**

By Eq. (22), we have

$$\|X_t - \overline{X}_t\|$$

$$= \left\|x + \int_0^t \frac{\mathrm{d}}{\mathrm{d}\tau} X_\tau \mathrm{d}\tau - x - \int_0^t \frac{\mathrm{d}}{\mathrm{d}\tau} \overline{X}_\tau \mathrm{d}\tau\right\|$$

$$\leq (1 - \alpha_t)\left\|\int_0^t \left(\boldsymbol{v}_{\mathrm{obs}}(X_\tau, \tau, c_{\mathrm{obs}}) - \bar{\boldsymbol{v}}_\theta(\overline{X}_\tau, \tau, c_{\mathrm{obs}})\right)\mathrm{d}\tau\right\| + \alpha_t\left\|\int_0^t \left(\boldsymbol{v}_{\mathrm{sem}}(X_t, t, c_{\mathrm{sem}}) - \bar{\boldsymbol{v}}_\theta(\overline{X}_\tau, \tau, c_{\mathrm{obs}})\right)\mathrm{d}\tau\right\|$$

$$\leq (1 - \alpha_t)\int_0^t \left\|\boldsymbol{v}_{\mathrm{obs}}(X_\tau, \tau, c_{\mathrm{obs}}) - \bar{\boldsymbol{v}}_\theta(\overline{X}_\tau, \tau, c_{\mathrm{obs}})\right\|\mathrm{d}\tau + \alpha_t\int_0^t \left\|\boldsymbol{v}_{\mathrm{sem}}(X_t, t, c_{\mathrm{sem}}) - \bar{\boldsymbol{v}}_\theta(\overline{X}_\tau, \tau, c_{\mathrm{obs}})\right\|\mathrm{d}\tau.$$

For the first term:

$$\int_0^t \left\| \boldsymbol{v}_{\mathrm{obs}}(X_\tau, \tau, c_{\mathrm{obs}}) - \bar{\boldsymbol{v}}_\theta(\overline{X}_\tau, \tau, c_{\mathrm{obs}}) \right\| \mathrm{d}\tau$$

$$\leq \int_0^t \left\| \boldsymbol{v}_{\mathrm{obs}}(X_\tau, \tau, c_{\mathrm{obs}}) - \bar{\boldsymbol{v}}_\theta(X_\tau, \tau, c_{\mathrm{obs}}) \right\| \mathrm{d}\tau + \int_0^t \left\| \bar{\boldsymbol{v}}_\theta(X_\tau, \tau, c_{\mathrm{obs}}) - \bar{\boldsymbol{v}}_\theta(\overline{X}_\tau, \tau, c_{\mathrm{obs}}) \right\| \mathrm{d}\tau$$

$$\leq \left( \int_0^t \left\| \boldsymbol{v}_{\mathrm{obs}}(X_\tau, \tau, c_{\mathrm{obs}}) - \bar{\boldsymbol{v}}_\theta(X_\tau, \tau, c_{\mathrm{obs}}) \right\|^2 \mathrm{d}\tau \right)^{1/2} + \int_0^t L_\tau^{\mathrm{nn}} \| X_\tau - \overline{X}_\tau \| \mathrm{d}\tau$$

(by Cauchy-Schwarz inequality and Lipschitz of $\bar{\boldsymbol{v}}_\theta$)

$$= \mathcal{E}_{\mathrm{obs}}(\bar{\boldsymbol{v}}_\theta) + \int_0^t L_\tau^{\mathrm{nn}} \| X_\tau - \overline{X}_\tau \| \mathrm{d}\tau. \qquad \text{(by the notation defined in Eq. (14))}$$

Similarly, for the second term, we have

$$\int_0^t \left\| \boldsymbol{v}_{\mathrm{sem}}(X_t, t, c_{\mathrm{sem}}) - \bar{\boldsymbol{v}}_\theta(\overline{X}_\tau, \tau, c_{\mathrm{obs}}) \right\| \mathrm{d}\tau$$

$$\leq \int_0^t \left\| \boldsymbol{v}_{\mathrm{sem}}(X_\tau, \tau, c_{\mathrm{sem}}) - \bar{\boldsymbol{v}}_\theta(X_\tau, \tau, c_{\mathrm{sem}}) \right\| \mathrm{d}\tau + \int_0^t \left\| \bar{\boldsymbol{v}}_\theta(X_\tau, \tau, c_{\mathrm{sem}}) - \bar{\boldsymbol{v}}_\theta(\overline{X}_\tau, \tau, c_{\mathrm{obs}}) \right\| \mathrm{d}\tau$$

$$\leq \left( \int_0^t \left\| \boldsymbol{v}_{\mathrm{sem}}(X_\tau, \tau, c_{\mathrm{obs}}) - \bar{\boldsymbol{v}}_\theta(X_\tau, \tau, c_{\mathrm{obs}}) \right\|^2 \mathrm{d}\tau \right)^{1/2} + \| c_{\mathrm{sem}} - c_{\mathrm{obs}} \| \int_0^t L_\tau^{\mathrm{nn}} \mathrm{d}\tau + \int_0^t L_\tau^{\mathrm{nn}} \| X_\tau - \overline{X}_\tau \| \mathrm{d}\tau$$

(by Cauchy-Schwarz inequality and Lipschitz of $\bar{\boldsymbol{v}}_\theta$)

$$= \mathcal{E}_{\mathrm{sem}}(\bar{\boldsymbol{v}}_\theta) + \| c_{\mathrm{sem}} - c_{\mathrm{obs}} \| \int_0^t L_\tau^{\mathrm{nn}} \mathrm{d}\tau + \int_0^t L_\tau^{\mathrm{nn}} \| X_\tau - \overline{X}_\tau \| \mathrm{d}\tau. \qquad \text{(by the notation defined in Eq. (15))}$$

Combining the above two inequalities, we obtain

$$\left\| X_t - \overline{X}_t \right\| \leq (1 - \alpha_t) \mathcal{E}_{\mathrm{obs}}(\bar{\boldsymbol{v}}_\theta) + \alpha_t \mathcal{E}_{\mathrm{sem}}(\bar{\boldsymbol{v}}_\theta) + \| c_{\mathrm{sem}} - c_{\mathrm{obs}} \| \int_0^t L_\tau^{\mathrm{nn}} \mathrm{d}\tau + \int_0^t L_\tau^{\mathrm{nn}} \| X_\tau - \overline{X}_\tau \| \mathrm{d}\tau.$$

Using Grönwall's inequality (see Theorem A.10), we get

$$\| X_t - \overline{X}_t \| \leq \left( (1 - \alpha_t) \mathcal{E}_{\mathrm{obs}}(\bar{\boldsymbol{v}}_\theta) + \alpha_t \mathcal{E}_{\mathrm{sem}}(\bar{\boldsymbol{v}}_\theta) + \| c_{\mathrm{sem}} - c_{\mathrm{obs}} \| \int_0^t L_\tau^{\mathrm{nn}} \mathrm{d}\tau \right) \exp\left( \int_0^t L_\tau^{\mathrm{nn}} \mathrm{d}\tau \right).$$

**(ii) bounding $\| X_t - \widehat{X}_t \|$**

Following the same derivations in (i), we have

$$\| X_t - \widehat{X}_t \| \leq (1 - \alpha_t) \int_0^t \left\| \boldsymbol{v}_{\mathrm{obs}}(X_\tau, \tau, c_{\mathrm{obs}}) - \bar{\boldsymbol{v}}_\theta(\widehat{X}_\tau, \tau, c_{\mathrm{obs}}) \right\| \mathrm{d}\tau + \alpha_t \int_0^t \left\| \boldsymbol{v}_{\mathrm{sem}}(X_t, t, c_{\mathrm{sem}}) - \tilde{\boldsymbol{v}}_\theta(\widehat{X}_\tau, \tau, c_{\mathrm{obs}}) \right\| \mathrm{d}\tau.$$

The first term can be upper-bounded as in (i). For the second term, following a similar derivation as in (i) for the second term, we obtain

$$\int_0^t \left\| \boldsymbol{v}_{\mathrm{sem}}(X_t, t, c_{\mathrm{sem}}) - \tilde{\boldsymbol{v}}_\theta(\widehat{X}_\tau, \tau, c_{\mathrm{obs}}) \right\| \mathrm{d}\tau$$

$$\leq \mathcal{E}_{\mathrm{sem}}(\tilde{\boldsymbol{v}}_\theta) + \| c_{\mathrm{sem}} - c_{\mathrm{prior}} \| \int_0^t \widetilde{L}_\tau^{\mathrm{nn}} \mathrm{d}\tau + \int_0^t \widetilde{L}_\tau^{\mathrm{nn}} \| X_\tau - \overline{X}_\tau \| \mathrm{d}\tau.$$

Combining these two inequalities, we obtain

$$\| X_t - \widehat{X}_t \| \leq (1 - \alpha_t) \mathcal{E}_{\mathrm{obs}}(\bar{\boldsymbol{v}}_\theta) + \alpha_t \mathcal{E}_{\mathrm{sem}}(\widetilde{\boldsymbol{v}}_\theta) + \| c_{\mathrm{sem}} - c_{\mathrm{prior}} \| \int_0^t \widetilde{L}_\tau^{\mathrm{nn}} \mathrm{d}\tau + \int_0^t \left( (1 - \alpha_t) L_\tau^{\mathrm{nn}} + \alpha_t \widetilde{L}_\tau^{\mathrm{nn}} \right) \| X_\tau - \overline{X}_\tau \| \mathrm{d}\tau.$$

Again, applying Grönwall's inequality, we get

$$\|X_t - \widehat{X}_t\| \leq \Big((1 - \alpha_t)\mathcal{E}_{\text{obs}}(\bar{\boldsymbol{v}}_\theta) + \alpha_t \mathcal{E}_{\text{sem}}(\widetilde{\boldsymbol{v}}_\theta) + \|c_{\text{sem}} - c_{\text{prior}}\| \int_0^t \widetilde{L}_\tau^{\text{nn}} \mathrm{d}\tau\Big) \exp\Big(\int_0^t \big((1 - \alpha_t)L_\tau^{\text{nn}} + \alpha_t \widetilde{L}_\tau^{\text{nn}}\big)\mathrm{d}\tau\Big).$$

$\square$

### A.3.2. WASSERSTEIN BOUNDS

**Theorem A.8.** *Let $T, \overline{T}, \widehat{T}$ be the transport maps induced by the ground truth flow Eq. (19), the neural network flow Eq. (20), and the Gaussian blur flow Eq. (21), respectively. Let $\rho = \mathcal{N}(0, \boldsymbol{I}_d)$. Let $\rho = \mathcal{N}(0, \boldsymbol{I}_d)$. Then the Wasserstein-2 distance between the generated distributions is bounded by:*

$$\mathsf{W}_2(T_\sharp \rho, \overline{T}_\sharp \rho) \leq \Big((1 - \alpha_{\max})\mathcal{E}_{\text{obs}}(\bar{\boldsymbol{v}}_\theta) + \alpha_{\max}\mathcal{E}_{\text{sem}}(\bar{\boldsymbol{v}}_\theta) + C_{\text{obs}}\Big) \exp\Big(\int_0^t L_\tau^{\text{nn}} \mathrm{d}\tau\Big),$$

$$\mathsf{W}_2(T_\sharp \rho, \widehat{T}_\sharp \rho) \leq \Big((1 - \alpha_{\max})\mathcal{E}_{\text{obs}}(\bar{\boldsymbol{v}}_\theta) + \alpha_{\max}\mathcal{E}_{\text{sem}}(\widetilde{\boldsymbol{v}}_\theta) + C_{\text{prior}}\Big) \exp\Big(\int_0^t \big((1 - \alpha_t)L_\tau^{\text{nn}} + \alpha_t \widetilde{L}_\tau^{\text{nn}}\big)\mathrm{d}\tau\Big).$$

*where $C_{\text{obs}} = \|c_{\text{sem}} - c_{\text{obs}}\| \int_0^t L_\tau^{\text{nn}} \mathrm{d}\tau$, and $C_{\text{prior}} = \|c_{\text{sem}} - c_{\text{prior}}\| \int_0^t \widetilde{L}_\tau^{\text{nn}} \mathrm{d}\tau$.*

*Proof.* By the definition of the Wasserstein distance, we have:

$$\mathsf{W}_2^2(T_\sharp \rho, \overline{T}_\sharp \rho) = \inf_{\gamma \in \Gamma(T_\sharp \rho, \overline{T}_\sharp \rho)} \int \|x - y\|^2 \mathrm{d}\gamma(x, y).$$

We construct a specific coupling $\gamma^*$ induced by the shared initialization. Let $z \sim \rho$ be the common latent variable. Let $x = T(z) = X_t$ and $y = \overline{T}(z) = \overline{X}_t$. The joint distribution of $(X_t, \overline{X}_t)$ forms a valid coupling in $\Gamma(T_\sharp \rho, \overline{T}_\sharp \rho)$.

Thus,

$$\mathsf{W}_2^2(T_\sharp \rho, \overline{T}_\sharp \rho) \leq \mathbb{E}_{z \sim \rho}\Big[\|T(z) - \overline{T}(z)\|^2\Big]$$
$$= \mathbb{E}_{z \sim \rho}\Big[\|X_t - \overline{X}_t\|^2\Big].$$

Using the Stability Lemma derived in Theorem A.7, we substitute the bound for $\|X_t - \overline{X}_t\|$. Let $K_t = \exp(\int_0^t L_\tau^{\text{nn}} \mathrm{d}\tau)$.

$$\mathbb{E}\Big[\|X_t - \overline{X}_t\|^2\Big] \leq K_t^2 \cdot \mathbb{E}\left[\left((1 - \alpha_t)\int_0^t \|\Delta\boldsymbol{v}_{\text{obs}}\|\mathrm{d}\tau + \alpha_t \int_0^t \|\Delta\boldsymbol{v}_{\text{sem}}\|\mathrm{d}\tau + C_{\text{sem}}\right)^2\right].$$

Applying the Minkowski inequality (triangle inequality for $L_2$ norm) to the expectation terms yields the final result in terms of $\mathcal{E}_{\text{obs}}$ and $\mathcal{E}_{\text{sem}}$:

$$\big(\mathbb{E}[\|X_t - \overline{X}_t\|^2]\big)^{1/2} \leq K_t\big((1 - \alpha_t)\mathcal{E}_{\text{obs}}(\bar{\boldsymbol{v}}_\theta) + \alpha_t \mathcal{E}_{\text{sem}}(\bar{\boldsymbol{v}}_\theta) + C_{\text{sem}}\big),$$

which gives

$$\mathsf{W}_2(T_\sharp \rho, \overline{T}_\sharp \rho) \leq \Big(\mathbb{E}_{z \sim \rho}\Big[\|X_t - \overline{X}_t\|^2\Big]\Big)^{1/2}$$
$$\leq \big((1 - \alpha_t)\mathcal{E}_{\text{obs}}(\bar{\boldsymbol{v}}_\theta) + \alpha_t \mathcal{E}_{\text{sem}}(\bar{\boldsymbol{v}}_\theta) + C_{\text{sem}}\big) \exp\big(\int_0^t L_\tau^{\text{nn}} \mathrm{d}\tau\big).$$

Follow the similar proof for $\mathsf{W}_2(T_\sharp \rho, \overline{T}_\sharp \rho)$, we can bound $\mathsf{W}_2(T_\sharp \rho, \widehat{T}_\sharp \rho)$. $\square$

**Theorem A.9** (Wasserstein Distance Bound). *Let $p$ be the distribution generated by the Ground Truth Flow, $\bar{p}$ by the Standard Flow, and $\hat{p}$ by the RelaxFlow. The Wasserstein-2 distance to the ground truth is bounded by:*

$$\mathcal{W}_2(p, \bar{p}) \leq C \cdot (\mathcal{E}_{\mathrm{obs}} + \mathcal{E}_{\mathrm{sem}}(\bar{\boldsymbol{v}}_\theta)), \tag{27}$$

$$\mathcal{W}_2(p, \hat{p}) \leq C \cdot (\mathcal{E}_{\mathrm{obs}} + \mathcal{E}_{\mathrm{sem}}(\tilde{\boldsymbol{v}}_\theta) + \delta_{\mathrm{prior}}), \tag{28}$$

*where $C$ is a constant depending on the Lipschitz continuity of the network (Grönwall factor), and $\delta_{\mathrm{prior}}$ accounts for the mismatch between $c_{\mathrm{sem}}$ and $c_{\mathrm{prior}}$.*

*Proof.* Let $\bar{p} = \overline{T}_\sharp \rho, \widehat{p} = \widehat{T}_\sharp \rho$ be the distribution generated by neural network flow Eq. (20) and the Gaussian blur flow Eq. (21), respectively. The proof follows identically to Theorem A.8 by choosing the coupling $(X_t, \widehat{X}_t)$ induced by $z \sim \rho$ and applying the second inequality from the Stability Lemma. □

Since $\mathcal{E}_{\mathrm{sem}}(\tilde{\boldsymbol{v}}_\theta) < \mathcal{E}_{\mathrm{sem}}(\bar{\boldsymbol{v}}_\theta)$ (due to the filtering of high-frequency noise), the Gaussian blur flow $\hat{p}$ achieves a tighter Wasserstein bound to the ground truth distribution $p$ than the standard flow $\bar{p}$, provided the conditioning mismatch $\delta_{\mathrm{prior}}$ is controlled.

**Lemma A.10** (Grönwall's Inequality). *Assume that the continuous functions $u, \kappa : [0, T] \to [0, \infty)$ and constant $K > 0$ satisfy:*

$$u(t) \leq K + \int_0^t \kappa(s)u(s)\mathrm{d}s,$$

*for all $t \in [0, T]$. Then:*

$$u(t) \leq K \exp\left(\int_0^t \kappa(s)\mathrm{d}s\right).$$

### A.4. Mechanism: Logit Smoothing as Vector Field Relaxation

Section 3.2 analyzes low-pass relaxation abstractly as an operator $\mathcal{R}_\sigma$ acting on the prior-conditioned velocity field (Eq. 2). In implementation, we realize this relaxation by blurring the prior-branch cross-attention logits before the softmax, which we now connect to an induced relaxation on the resulting velocity.

Let $v_\theta(x, t, c)$ denote the rectified-flow velocity produced by the transformer, and let $L(x, t, c)$ denote the corresponding cross-attention logits inside the network. In the Semantic-Prior Branch, we replace $L$ by $\tilde{L} = G_\sigma * L$ (Eq. 8) and denote the induced velocity by $\tilde{v}_\theta(x, t, c_{\mathrm{prior}}) := v_\theta(x, t, c_{\mathrm{prior}}; \tilde{L})$.

**Assumption A.11** (Logit-to-velocity smoothness). Assume the velocity is Lipschitz in the logits with constant $C_t$:

$$\|v_\theta(x, t, c; L_1) - v_\theta(x, t, c; L_2)\| \leq C_t \|L_1 - L_2\|.$$

Under Assumption A.11 and the fact that Gaussian blurring contracts high-frequency logit components, we obtain

$$\|\tilde{v}_\theta(x, t, c_{\mathrm{prior}}) - v_\theta(x, t, c_{\mathrm{prior}})\| \leq C_t \|\tilde{L} - L\|,$$

showing that logit smoothing induces a controlled relaxation of the prior-conditioned velocity.

## B. Dataset Curation

### B.1. ExtremeOcc-3D Dataset

**Data source.** We build ExtremeOcc-3D from 3D-FUTURE (Fu et al., 2021b) and 3D-FRONT (Fu et al., 2021a), which provide indoor scenes with realistic furniture arrangements and high-quality ground-truth 3D objects. The rendered images capture natural viewpoints where occlusions and ambiguity arise organically.

**Occlusion ratio computation.** Using ground-truth meshes and poses, we reproject each object's amodal silhouette onto the scene image and compare it with the visible mask to compute the occlusion ratio $r$. Empirically, when $r \geq 80\%$, humans struggle to identify the object type from visible evidence alone.

**Failure modes under extreme occlusion.** State-of-the-art occlusion-aware models (*e.g.*, SAM3D (Chen et al., 2025a)) typically fail under such conditions in two ways: (1) generating incomplete, implausible geometry, or (2) producing a semantically incorrect but visually plausible object (*e.g.*, generating a chair for an occluded bed in a bedroom scene).

**Final dataset.** We filter approximately 2000 candidates with $r \geq 80\%$ and obtain 264 test cases. Each case is paired with a category-level text prior. Ground-truth 3D meshes are provided for evaluation. Fig. 8 shows examples of the ExtremeOcc-3D dataset.

### B.2. AmbiSem-3D Dataset

**Motivation.** In many real-world 3D generation settings—such as single-view reconstruction, novel view synthesis, or partial observation—the input image does not uniquely determine the underlying 3D structure or identity. Humans naturally rely on language and prior knowledge to resolve such ambiguity. However, existing benchmarks rarely test whether text guidance actually resolves semantic uncertainty, rather than merely refining appearance or style.

AmbiSem-3D is a diagnostic benchmark designed to evaluate text-guided 3D generation under incomplete or ambiguous visual observation. Unlike conventional 3D datasets that assume a single correct interpretation for each input, AmbiSem-3D focuses on semantic branching scenarios, where a single visual input admits multiple plausible but mutually exclusive semantic explanations.

**Curation process.** Three annotators manually reviewed rendered images from a high-quality diverse 3D object dataset, ObjaverseXL (Deitke et al., 2023), and carefully designed 21 cases exhibiting one of three ambiguity types:

- **Masked semantic completion:** key semantic regions are occluded.

- **View-induced identity ambiguity:** limited viewpoints prevent unique identification.

- **Intrinsic shape ambiguity:** geometry admits multiple semantic interpretations.

**Distinction from training data.** These cases differ fundamentally from typical training data: we use single rendered images from ambiguous viewpoints or under heavy occlusion, paired with text priors leading to diverse target objects distinct from the original 3D assets. This tests whether models can overcome their learned biases and generate diverse, branch-consistent outputs.

**Dataset structure.** Each case consists of: (1) a single ambiguous input image, and (2) a set of text branches, where each branch represents a distinct but plausible semantic interpretation. All branches are valid given the visual evidence, yet correspond to different 3D structures or categories. Fig. 7 shows all samples of the AmbiSem-3D dataset. Qualitative results are shown in Fig. 4, Fig. 12, Fig. 13, and Fig. 14.

**100-case extension.** For the extended evaluation in Appendix E, we construct a larger AmbiSem-3D-Ext split using a semi-automatic candidate mining pipeline. We first render ObjaverseXL assets from multiple viewpoints and caption each view with a VLM. A second VLM/LLM agent then compares the captions across views and identifies cases where one view yields a substantially different semantic interpretation from the consensus of the remaining views. Such views are treated as candidate ambiguous observations, since their visible evidence may support multiple plausible completions. The automatic stage mines 120 candidate cases, which we then manually filter to remove 20 clearly unreasonable samples, such as cases with insufficient asset quality or implausible ambiguity. The resulting extension contains 100 retained cases and is intended to broaden the evaluation toward view-induced ambiguity, complementing the inherent ambiguities emphasized by the 21-case diagnostic set.

## C. Prior Image Construction

Our method requires prior images to provide semantic guidance. We describe how priors are obtained for each benchmark during our experiments.

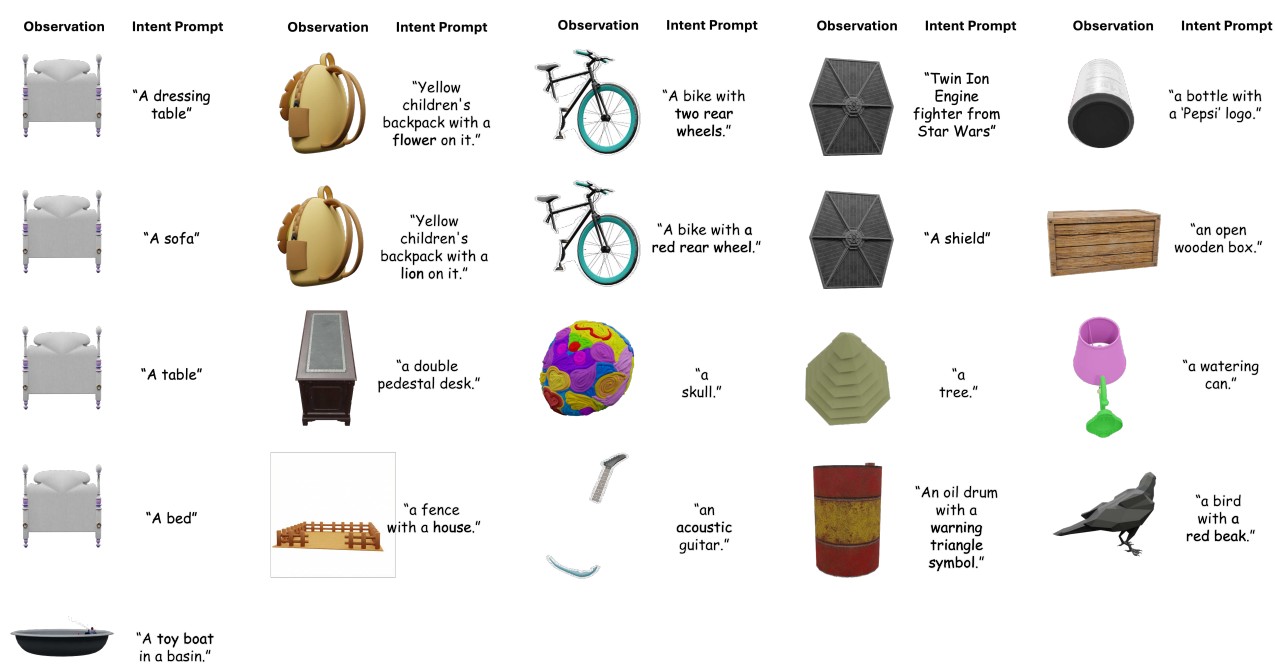

*Figure 7.* Visualization of all cases in AmbiSem-3D.

**ExtremeOcc-3D.** Given a text label specifying the target category, we retrieve a prior image by randomly sampling a *different* object of the same category from the 3D-FUTURE dataset and rendering it from a canonical viewpoint. This retrieval-based approach ensures that the prior shares the intended category but differs in instance-specific details, testing whether our method extracts only category-level semantics.

**AmbiSem-3D.** Since AmbiSem-3D involves diverse semantic branches without a retrieval pool, we generate prior images from text prompts using a text-to-image generator Z-Image (Cai et al., 2025). To ensure clean, 3D-compatible outputs, we augment the user prompt with a suffix: "`, clean 3D style, pure white background, single object in the center.`" For fair comparison, all baseline methods on AmbiSem-3D receive the same generated prior images as input.

**Ablation on prior source.** To verify robustness to prior source, we also evaluate RelaxFlow on ExtremeOcc-3D using Z-Image-generated priors (Cai et al., 2025) instead of retrieved images. As shown in Table 6a ("Prior from generation"), performance remains competitive (Point-FID: 82.7 vs. 81.1), indicating that our method is not sensitive to whether priors are retrieved or generated.

## D. Details of Metric Computation

We report a set of complementary metrics that capture semantic alignment to the text prior, semantic/appearance agreement with reference renders, fidelity at the distribution level, and 3D semantic similarity. For each test instance, our method produces $V=10$ predicted views $\mathcal{V}_{\text{pred}} = \{v_i\}_{i=1}^{V}$, together with a prior text prompt $t$. We also consider a set of processed ground-truth multi-view renders $\mathcal{V}_{\text{gt}}^{\text{proc}}$ and an observed object render $o^{\text{proc}}$ (from a specific viewpoint). When a metric requires standardized inputs, we apply a consistent preprocessing pipeline (center-crop and resize), yielding processed predicted views $\mathcal{V}_{\text{pred}}^{\text{proc}} = \{\tilde{v}_i\}_{i=1}^{V}$.

**CLIP image–text similarity.** We measure how well the predicted views align with the textual prior using CLIP image–text cosine similarity. Let $f_I(\cdot)$ and $f_T(\cdot)$ denote the CLIP image and text encoders, and $\hat{x} = x/\|x\|_2$ the $\ell_2$-normalized

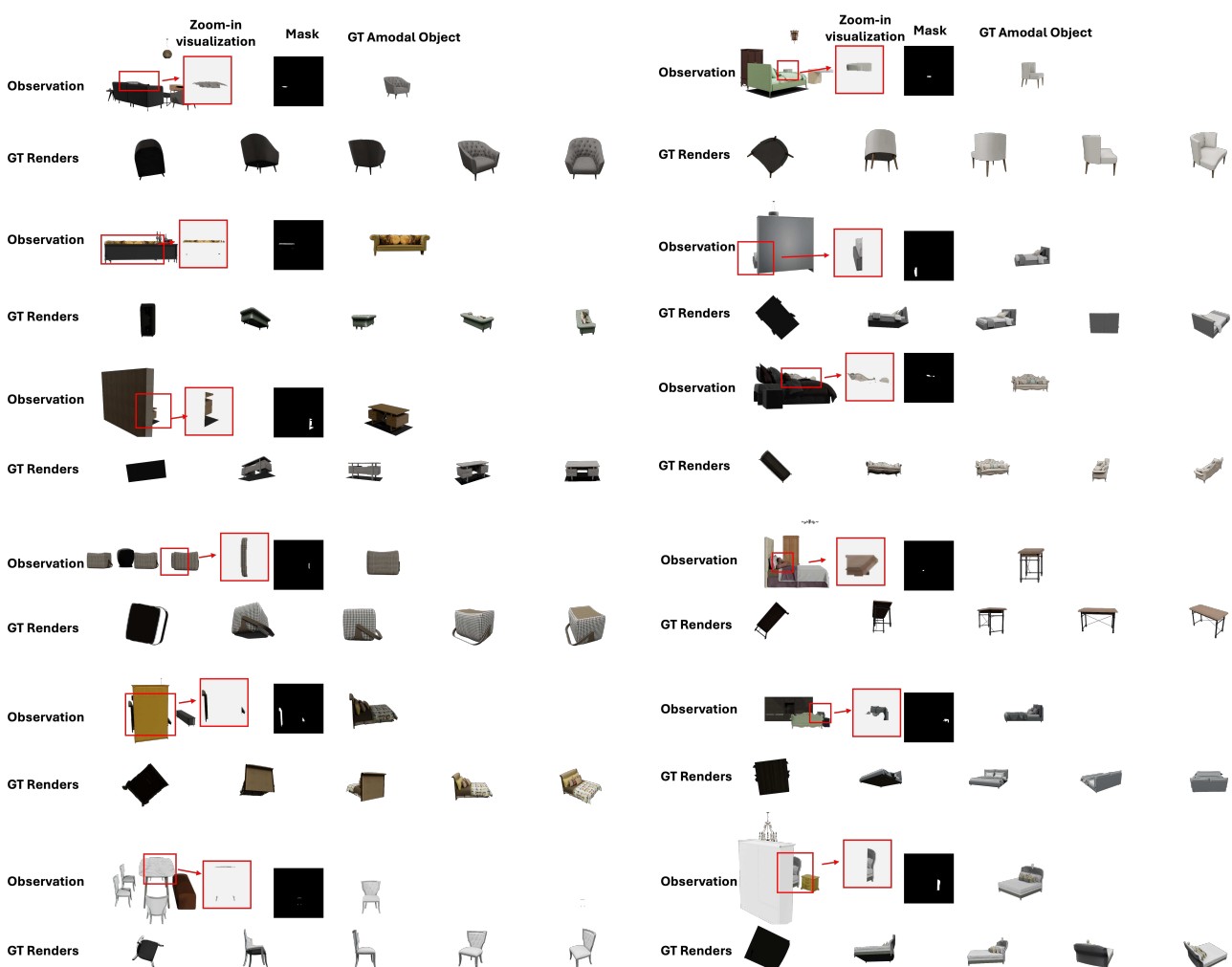

*Figure 8.* More examples of cases in ExtremeOcc-3D.

embedding. We compute

$$s_{\text{IT}} = \frac{1}{V} \sum_{i=1}^{V} \left\langle \widehat{f_I(v_i)}, \widehat{f_T(t)} \right\rangle, \tag{29}$$

where higher values indicate stronger semantic agreement between the generated imagery and the text prompt.

**CLIP image–image similarity (multi-view semantic agreement).**   To quantify overall agreement between the predicted object and the reference object at the semantic level, we compare the predicted views against a set of reference images that aggregates ground-truth renders and the observed render:

$$\mathcal{R} = \mathcal{V}_{\text{gt}}^{\text{proc}} \cup \{o^{\text{proc}}\}. \tag{30}$$

We embed each image with the CLIP image encoder, form the mean reference embedding, and average cosine similarities over predicted views:

$$\mu_{\mathcal{R}} = \frac{1}{|\mathcal{R}|} \sum_{r \in \mathcal{R}} \widehat{f_I(r)}, \qquad s_{\text{II}} = \frac{1}{V} \sum_{i=1}^{V} \left\langle \widehat{f_I(\tilde{v}_i)}, \widehat{\mu_{\mathcal{R}}} \right\rangle. \tag{31}$$

This metric summarizes *multi-view* consistency at a semantic representation level, reflecting whether the predicted set depicts the same underlying object content as the reference set (higher is better).

**Minimum LPIPS distance (single-view appearance consistency).**   While CLIP image–image similarity emphasizes *set-level, multi-view* semantic agreement, we additionally evaluate how well the method preserves the *observed* appearance from its specific viewpoint using LPIPS. Let $d_{\text{LPIPS}}(\cdot, \cdot)$ denote the LPIPS perceptual distance. We compute the minimum distance between the observed render and the predicted views:

$$d_{\min} = \min_{i \in \{1, \dots, V\}} d_{\text{LPIPS}}(o^{\text{proc}}, \tilde{v}_i). \tag{32}$$

Lower $d_{\min}$ indicates that at least one predicted view closely matches the observed image in perceptual appearance, which is particularly relevant because the observation provides supervision only at that single viewpoint.

**FID on multi-view image sets.**   We also report Fréchet Inception Distance (FID) (Heusel et al., 2017b) to assess distribution-level realism and coverage of the predicted multi-view images relative to ground-truth renders. For each object, we treat its $V$ predicted views as samples contributing to an overall generated set, and analogously aggregate ground-truth views into a reference set. Using Inception features $\phi(\cdot)$, we fit Gaussians to the feature distributions of the generated and reference sets, with means and covariances $(m_g, C_g)$ and $(m_r, C_r)$, respectively. The FID is

$$\text{FID} = \|m_g - m_r\|_2^2 + \text{Tr}\left( C_g + C_r - 2\left( C_g C_r \right)^{1/2} \right), \tag{33}$$

where lower values indicate closer alignment between the generated and ground-truth image distributions.

**Point-FID (3D set-level semantic similarity).**   To complement image-based fidelity with a 3D semantic metric, we report Point-FID computed from point-cloud features extracted by Point-E (Nichol et al., 2022). For each object, we obtain a point cloud from the predicted 3D output and a point cloud from the ground-truth 3D shape, then compute Point-E feature embeddings $\psi(\cdot)$. As with FID, we aggregate embeddings across the dataset into generated and reference distributions, estimate Gaussian statistics $(\tilde{m}_g, \tilde{C}_g)$ and $(\tilde{m}_r, \tilde{C}_r)$, and compute

$$\text{Point-FID} = \|\tilde{m}_g - \tilde{m}_r\|_2^2 + \text{Tr}\left( \tilde{C}_g + \tilde{C}_r - 2\left( \tilde{C}_g \tilde{C}_r \right)^{1/2} \right). \tag{34}$$

Lower Point-FID indicates that the predicted 3D shapes are closer to ground truth in the learned 3D semantic feature space, providing a set-level measure of 3D semantic similarity beyond per-view image agreement.

# E. More Baseline Comparisons and Efficiency Details

**Other baseline families.** We evaluate several alternative routes for injecting semantic intent into image-to-3D generation: video generation followed by 3D generation, multi-view generation followed by 3D generation, commercial 2D editing followed by 3D generation, and direct text+image conditioning. The video-generation baseline uses Wan2.2-TI2V (Wan et al., 2025) to generate orbit-style frames from the observation and intent prompt, followed by TRELLIS multi-image reconstruction (Xiang et al., 2025). The multi-view-generation baseline uses MV-Adapter (Huang et al., 2025) with the same reconstruction protocol. The commercial 2D-editing baseline replaces SDXL with Nano Banana Pro (Raisinghani, 2025) before 3D generation. The direct text+image baseline uses TRELLIS with an image-conditioned observation branch and a text-conditioned semantic branch combined by CFG-style interpolation.

Tables 3 and 4 show that these alternatives do not jointly preserve the observation and maintain 3D consistency. In particular, RelaxFlow achieves the best $CLIP_{img}$ and Point-FID on ExtremeOcc-3D and the best $CLIP_{img}$ on AmbiSem-3D, while maintaining competitive $CLIP_{txt}$. Figures 9, 10, and 11 further illustrate the failure modes: direct text+image fusion can misalign branches, video/multi-view generation can produce inconsistent reconstruction evidence, and non-3D-aware commercial editing can introduce geometry artifacts or attribute leakage.

*Table 3.* **Comparison with other baseline families on ExtremeOcc-3D.** These baselines cover video generation, multi-view generation, direct text+image conditioning, and commercial 2D editing pipelines.

| Method | $CLIP_{img}\uparrow$ | $CLIP_{txt}\uparrow$ | FID$\downarrow$ | LPIPS$\downarrow$ | Point-FID$\downarrow$ |
|---|---|---|---|---|---|
| Wan2.2-TI2V + TRELLIS | 0.78 | 22.98 | 107.49 | 0.85 | 110.3 |
| MV-Adapter + TRELLIS | 0.80 | 22.70 | 148.66 | 0.81 | 238.3 |
| TRELLIS (text+image) | 0.82 | 26.36 | 116.51 | 0.79 | 137.7 |
| Nano Banana Pro + 3D | 0.83 | **27.49** | 77.68 | 0.77 | 88.6 |
| **Ours** | **0.87** | 27.26 | **39.44** | **0.51** | **81.1** |

*Table 4.* **Comparison with other baseline families on AmbiSem-3D.** Because AmbiSem-3D has multiple valid completions and no unique 3D ground truth, we report CLIP-based observation and text alignment.

| Method | $CLIP_{img}\uparrow$ | $CLIP_{txt}\uparrow$ |
|---|---|---|
| Wan2.2-TI2V + TRELLIS | 0.80 | 26.71 |
| MV-Adapter + TRELLIS | 0.81 | 25.54 |
| TRELLIS (text+image) | 0.81 | 27.24 |
| Nano Banana Pro + 3D | 0.81 | **27.41** |
| **Ours** | **0.87** | 27.23 |

**Extended ambiguity evaluation.** We evaluate a 100-case extension of AmbiSem-3D (details are in Appendix B). The original 21-case set focuses on inherent ambiguity, while the extended set also includes view-induced ambiguity. As shown in Table 5, the trend remains consistent: RelaxFlow improves both observation and prompt alignment over SAM3D.

*Table 5.* **Extended 100-case AmbiSem-3D evaluation.**

| Method | $CLIP_{img}\uparrow$ | $CLIP_{txt}\uparrow$ |
|---|---|---|
| SAM3D | 0.839 | 25.6 |
| **Ours** | **0.845** | **26.9** |

**Runtime and memory.** We measure inference cost on a 10-sample random subset using a single NVIDIA RTX A5000 GPU. The results are shown in Table 6. RelaxFlow adds extra branch evaluations, increasing runtime, while peak memory grows modestly. The current implementation prioritizes memory over runtime; batched branch inference and feature caching may further reduce wall-clock overhead.

# F. Visibility Mask Details and Hyperparameters

This appendix provides the concrete parameterization used to compute the visibility-aware blending mask in Sec. 4.3.

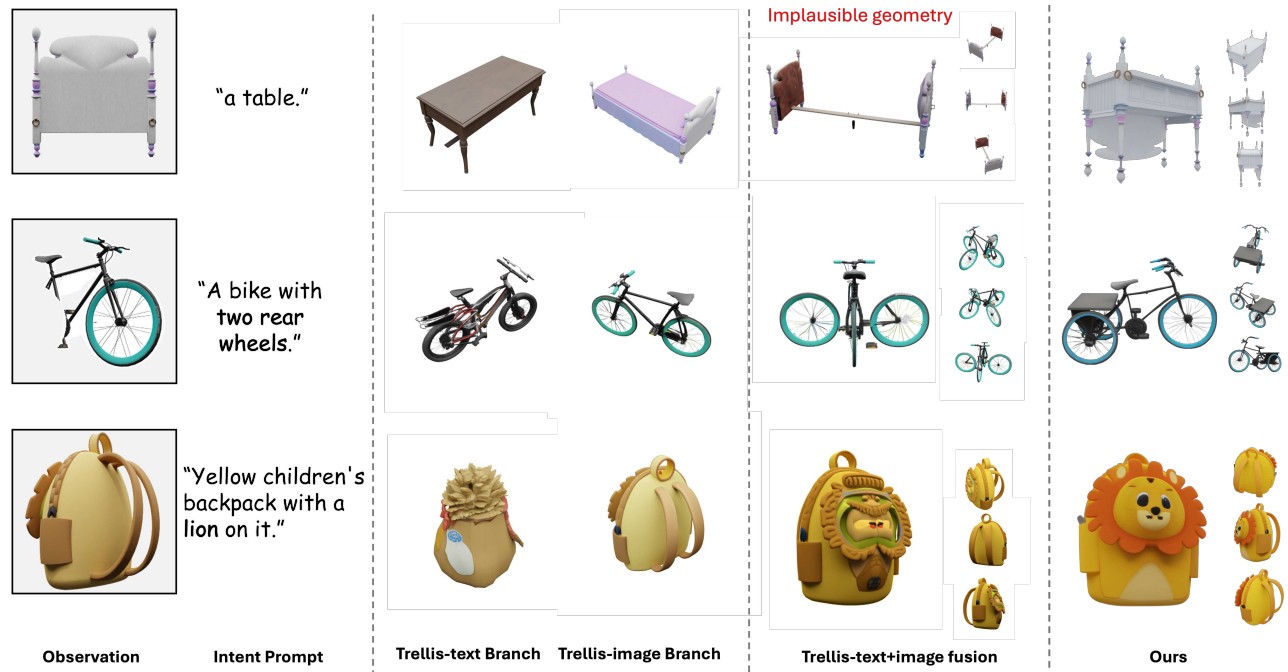

*Figure 9.* Qualitative comparison with the TRELLIS (text+image) fusion baseline. This baseline often produces implausible geometry because the objects generated by its text- and image-conditioned branches are not visually consistent, making spatial alignment difficult. In contrast, our method injects text conditioning as low-pass semantic guidance in the visual latent space, yielding more coherent and geometrically consistent results.

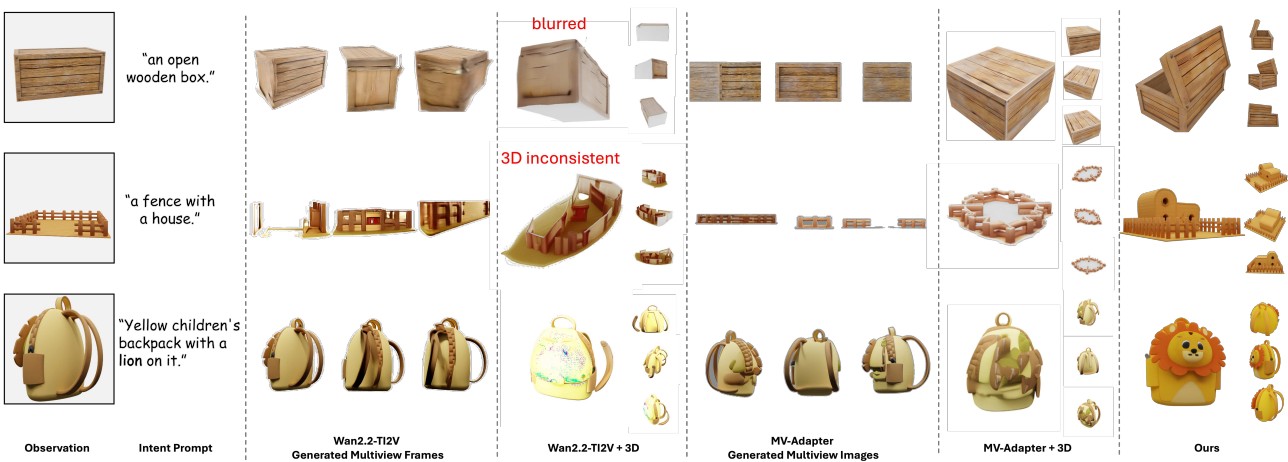

*Figure 10.* Qualitative comparison with Wan2.2-TI2V + TRELLIS and MV-Adapter + TRELLIS. For Wan2.2-TI2V, we use the enhanced prompt: "A clean studio turnaround video of intent prompt, centered in frame, isolated from the background, with the camera orbiting smoothly around the object to reveal multiple viewpoints. The object stays fixed while only the camera moves. Consistent geometry, consistent appearance, product-shot lighting, plain neutral background, the full object remains visible in every frame, no extra objects, no scene changes, no cuts, no zoom." Despite this careful prompting, the video-generation pipeline often fails to produce 3D-consistent frames, leading to distorted or blurred 3D reconstructions. The multi-view generation pipeline also struggles in this setting, often failing to preserve the observation evidence while following the text prompt, which results in distorted outputs. In contrast, our method operates during 3D generation and produces results that are both geometrically consistent and faithful to the input observation and text intent.

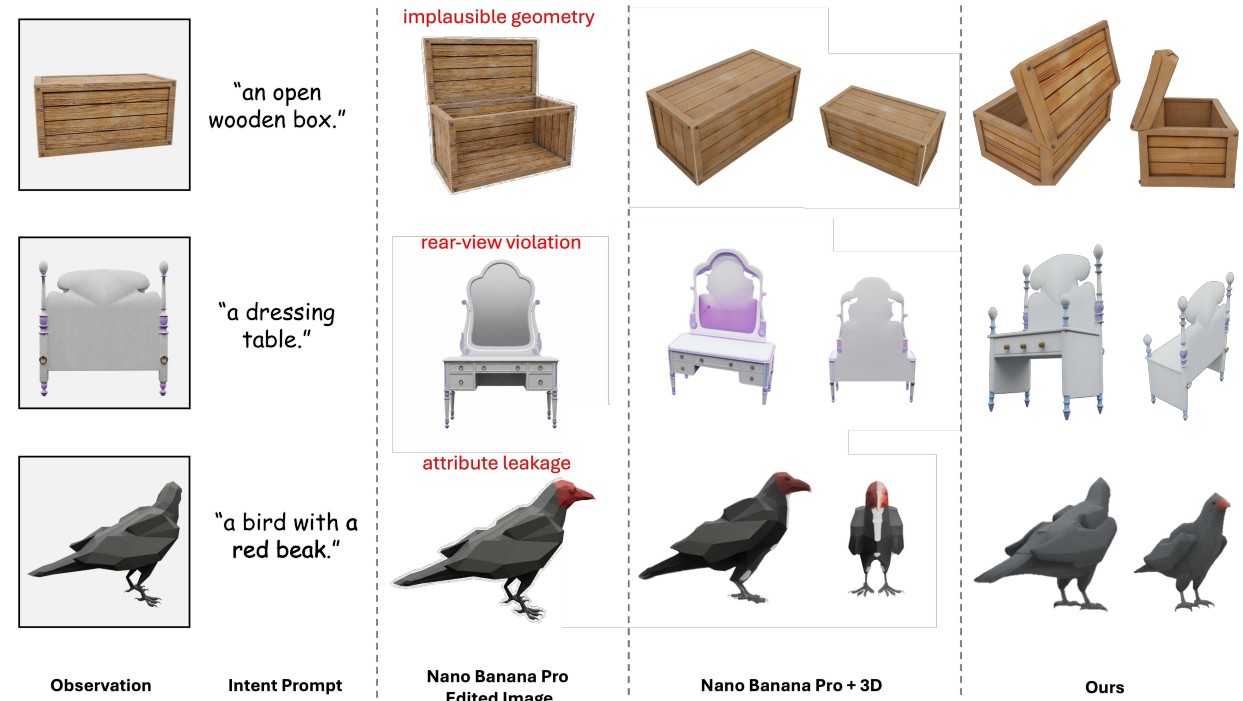

*Figure 11.* Qualitative comparison with Nano Banana Pro + 3D. As a non-3D-aware commercial 2D editing model, Nano Banana Pro can produce edits with implausible geometry, back-view inconsistency, and attribute leakage. In contrast, our method achieves text-guided amodal 3D generation that remains consistent with both the text intent and the observation evidence.

*Table 6.* **Runtime and memory overhead.** Memory is peak GPU memory in GB. RelaxFlow adds extra branch evaluations, increasing runtime, while peak memory grows modestly. The current implementation prioritizes memory over runtime; batched branch inference and feature caching may further reduce wall-clock overhead.

| Backbone | Method | Time (s/sample) | Allocated (GB) | Reserved (GB) |
|---|---|---|---|---|
| | Base | 15.59 | 18.07 | 21.85 |
| SAM3D | Ours | 38.40 | 18.60 | 22.60 |
| | Overhead | 2.46× | +2.9% | +3.4% |
| | Base | 12.81 | 5.57 | 5.60 |
| TRELLIS | Ours | 23.31 | 6.02 | 6.07 |
| | Overhead | 1.82× | +8.0% | +8.4% |

**Depth map construction.** Given the pose predicted by the backbone under $c_{\mathrm{obs}}$, we transform voxel indices $c_i \in \{0, \ldots, 63\}^3$ to camera space

$$x_i = s \odot (Rc_i) + t, \qquad x_i = (x_i, y_i, z_i), \tag{35}$$

and project to pixels $(u_i, v_i)$ with standard pinhole intrinsics $K = (f_x, f_y, c_x, c_y)$:

$$u_i = c_x - f_x \frac{x_i}{z_i}, \quad v_i = c_y - f_y \frac{y_i}{z_i}. \tag{36}$$

We build a z-buffer depth map

$$D(u, v) = \min\{ z_i \mid (u_i, v_i) = (u, v) \}, \tag{37}$$

and densify it by min-pooling dilation with kernel size $k$ (odd integer), producing $D'$.

**Voxel-aware scaling.** Let $s_{\max}$ denote the maximum component of the estimated scale $s$, and $s_{\mathrm{res}}$ the voxel resolution, we can define the object-space voxel size $s_{\mathrm{vox}} = s_{\max}/s_{\mathrm{res}}$.

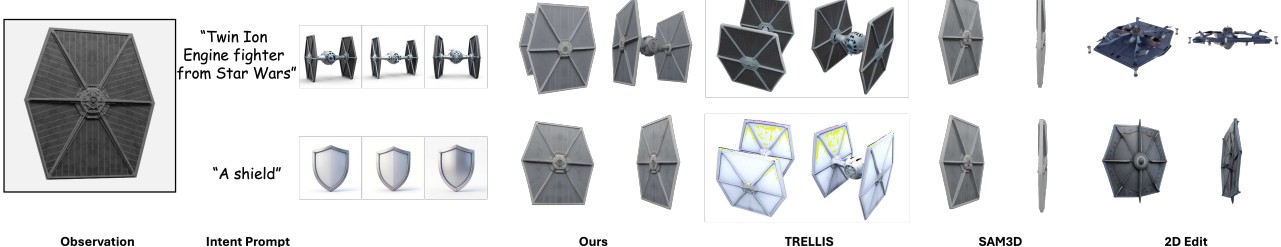

*Figure 12.* More qualitative results. TRELLIS and SAM3D remain overfitted to the observed geometry, producing either a shield or a TIE fighter regardless of intent, whereas our method generates either structure as specified by the intent prompt.

We further denote the object depth by $z_{\text{obj}} = t_z$ and the average focal length by $f_{\text{avg}} = \frac{f_x + f_y}{2}$. We set the depth tolerance to scale with voxel size:

$$\sigma_d = \beta\, s_{\text{vox}}, \tag{38}$$

and choose the dilation kernel based on the approximate projected voxel footprint:

$$k = \text{odd}\left(\text{round}\left(\gamma\, s_{\text{vox}}\, \frac{f_{\text{avg}}}{z_{\text{obj}}}\right)\right), \tag{39}$$

where $\text{odd}(\cdot)$ rounds to the nearest odd integer (and we clamp to $k \geq 1$ in implementation). Intuitively, $s_{\text{vox}} \frac{f_{\text{avg}}}{z_{\text{obj}}}$ approximates how many pixels a voxel spans under perspective projection, so $k$ adapts to both object scale and depth.

**Soft visibility weight.** For each voxel $i$, with projected pixel $(u_i, v_i)$ and depth $z_i$, we compute the occlusion margin

$$\Delta_i = z_i - D'(u_i, v_i), \tag{40}$$

and convert it into a soft visibility weight

$$m_i = \exp\left(-\lambda \left(\frac{\max(\Delta_i, 0)}{\sigma_d}\right)^2\right). \tag{41}$$

**Default values.** Unless otherwise specified, we use fixed defaults across all experiments:

$$\beta = 1.5, \qquad \gamma = 1.5, \qquad \lambda = 3. \tag{42}$$

These settings make the visibility boundary stable under discretization and modest pose noise while remaining sufficiently selective to prevent prior overpainting on visible surfaces.

## G. More Qualitative Results

See Fig. 12, Fig. 14, Fig. 13, and Fig. 15.

## H. Failure Cases

Our method assumes that the observation and prior are broadly compatible: the observation is ambiguous in some aspects, and the prior supplies complementary cues to resolve that ambiguity. When the two inputs contain mutually conflicting evidence (*e.g.*, two different frontal faces with incompatible identities), the model has no consistent solution and may fail to preserve either condition faithfully. In these cases, our method tends to act as a soft "concept fusion" mechanism, interpolating between the two conditions due to their asymmetric guidance. We illustrate this behavior on cases from the morphing benchmark from Interp3D (Liu et al., 2026) (see Fig. 16).

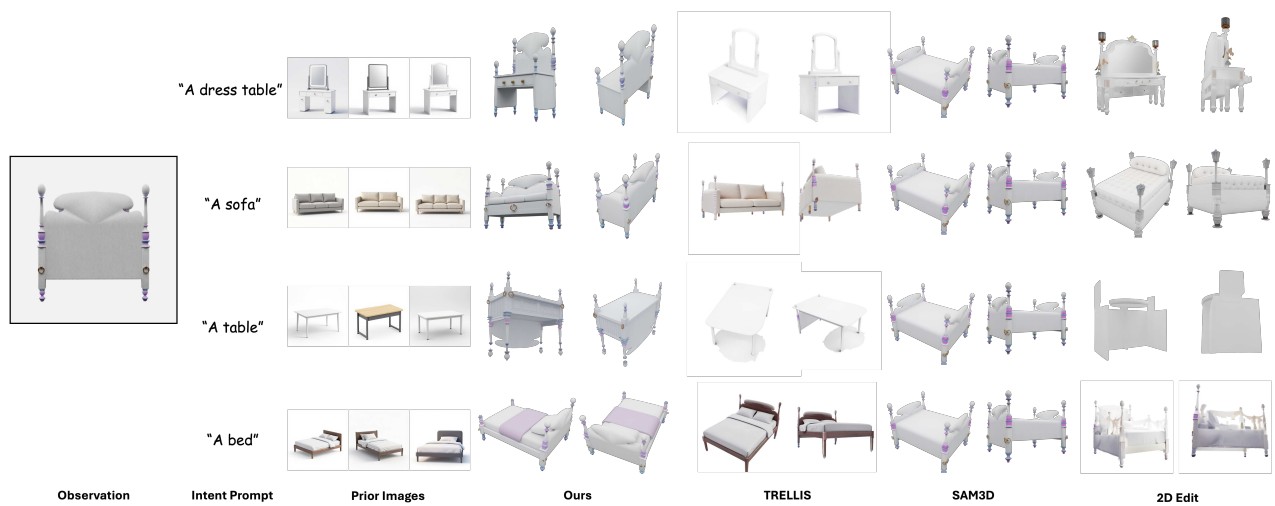

*Figure 13.* More qualitative results.

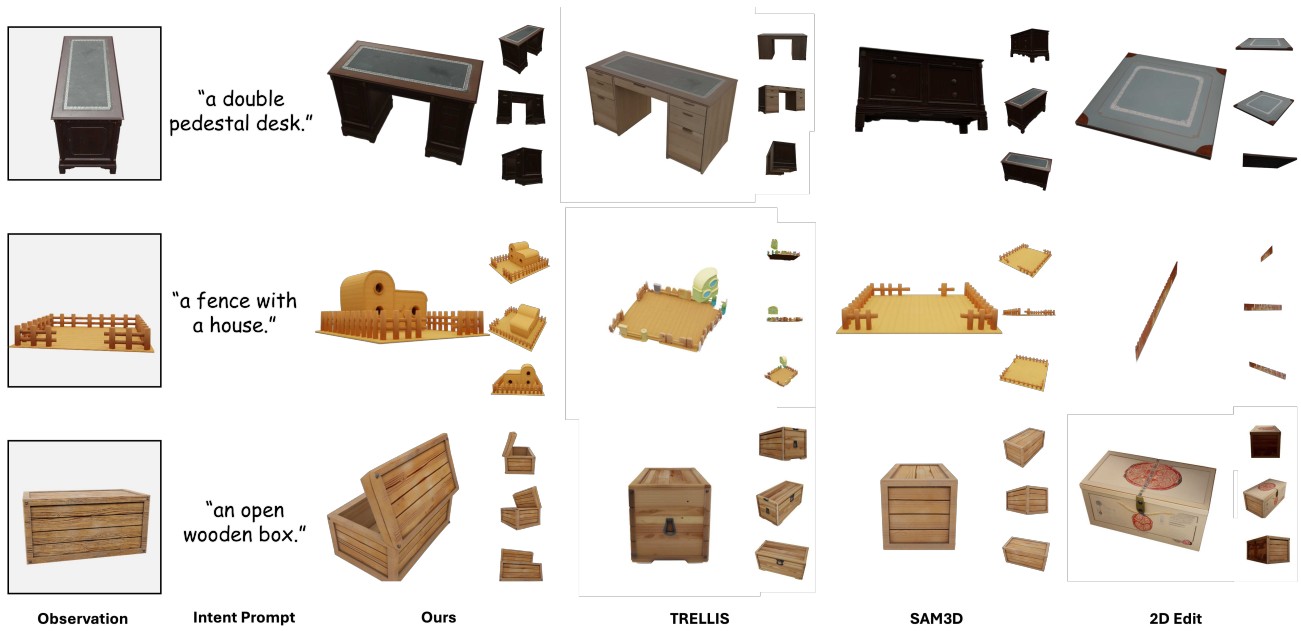

*Figure 14.* More qualitative results.

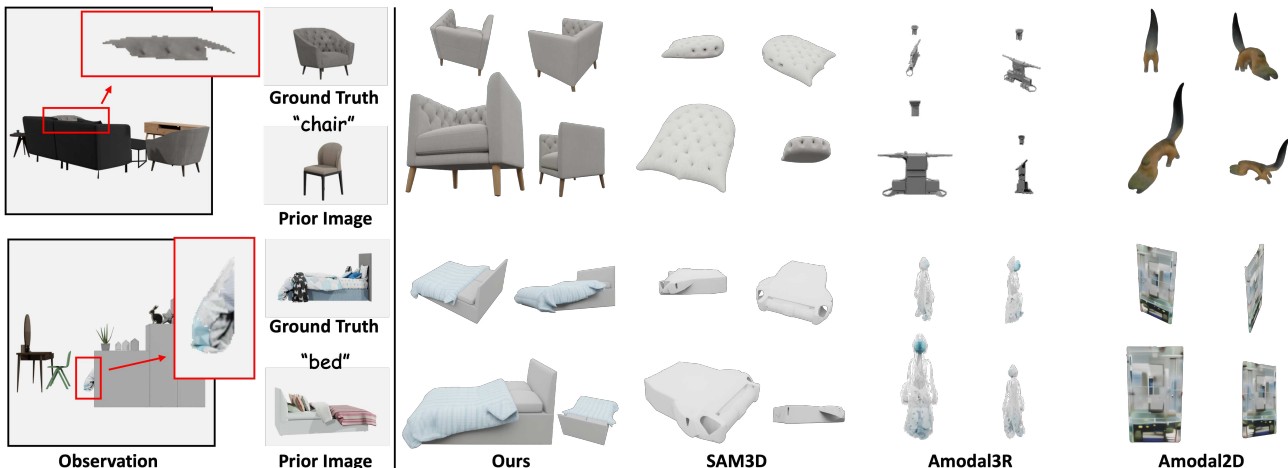

*Figure 15.* More qualitative results on ExtremeOcc-3D.

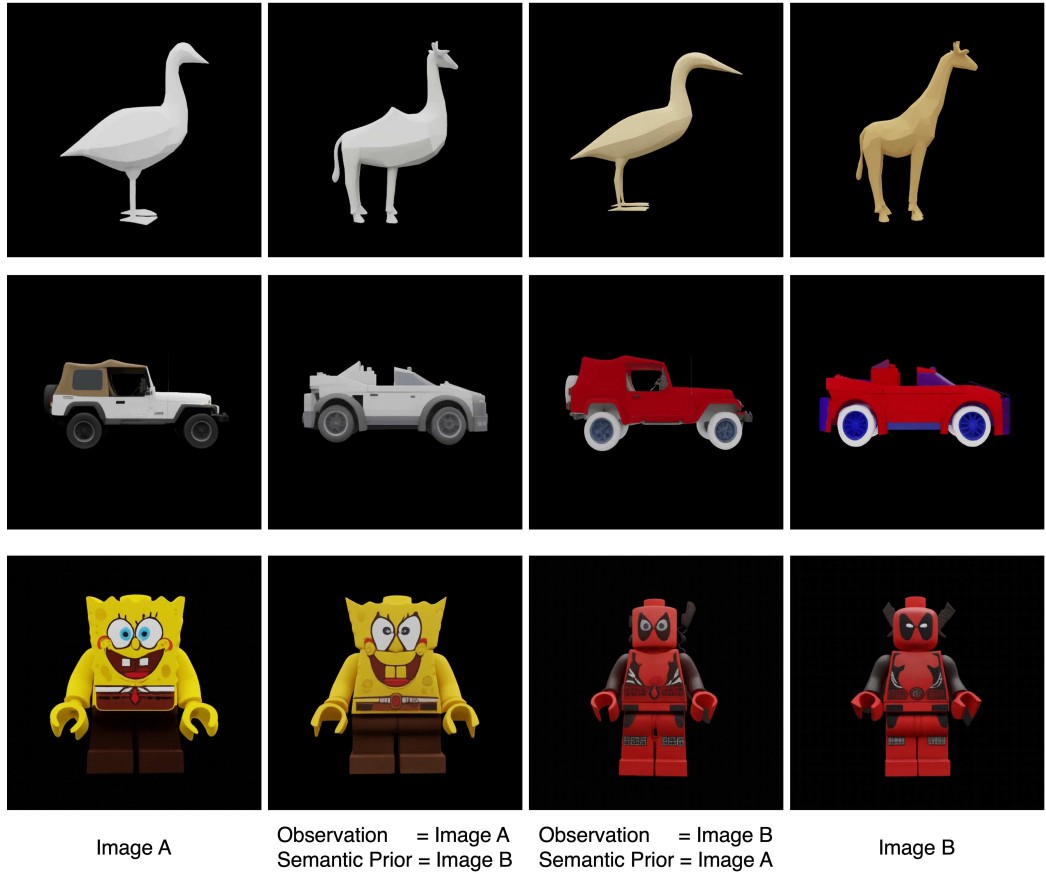

*Figure 16.* Failure cases under conflicting conditions. When the observation and prior disagree, no evidence-consistent solution exists, and our method may interpolate between them as a soft concept fusion due to asymmetric guidance.

