# OpenReview forum: "RelaxFlow: Text-Driven Amodal 3D Generation"
_ICML.cc/2026/Conference — ICML 2026 spotlight_

### Official Review · Reviewer_JzMo · 2026-02-28

**Soundness:** 3
**Presentation:** 3
**Significance:** 3
**Originality:** 3
**Overall Recommendation:** 4
**Confidence:** 4

**Summary:**

This paper proposes RelaxFlow, a training-free dual-branch inference framework designed to mitigate semantic ambiguity in image-to-3D generation under severe occlusion. The method integrates observation-consistent guidance with a text-driven multi-prior consensus mechanism, aiming to preserve visible evidence while enabling semantically plausible amodal completion. Experiments on challenging occlusion benchmarks show improved semantic alignment compared to recent feed-forward baselines.

**Compliance With Llm Reviewing Policy:**

Affirmed.

**Final Justification:**

The paper presents a practical training-free framework for ambiguity-aware amodal image-to-3D generation. My main concerns were the very small AmbiSem-3D benchmark, the incomplete transfer to TRELLIS because the visibility-aware mask is removed there, and the lack of direct 3D evaluation on hidden regions. In the rebuttal, the authors added confidence intervals and significance tests, reported ablations with and without the visibility-aware mask, and included hidden-region 3D metrics. These additions strengthen the paper and partially address my concerns, but the ambiguity benchmark is still small and the cross-backbone story remains somewhat incomplete.

I maintain my score.

**Key Questions For Authors:**

1. In the TRELLIS integration, the visibility-aware mask is removed due to missing pose estimation. How much performance is lost compared to the full RelaxFlow design? Can the authors provide ablations demonstrating the necessity of the visibility mechanism on backbones where it is available?

2. AmbiSem-3D contains only 21 cases. Can the authors report confidence intervals or statistical significance tests for the human preference and quantitative metrics? Are there plans or preliminary results on a larger-scale ambiguity benchmark?

3. Since the core claim concerns amodal geometric reasoning, can the authors provide region-wise 3D metrics (e.g., Chamfer or normal error on occluded regions separately from visible regions) to rule out improvements that are primarily perceptual?

**Limitations:**

Partially. The paper discusses some practical constraints, but the statistical limitation of AmbiSem-3D and the dependency on prior quality should be more explicitly acknowledged.

**Strengths And Weaknesses:**

**Strengths**

The paper tackles the highly relevant problem of ambiguity-aware amodal 3D generation with a well-motivated approach. The core system-level contribution, combining observation-consistent flow with text-driven multi-prior consensus, is structurally novel and offers a practical training-free solution that can plug into existing pipelines. The methodology is generally easy to follow, and the authors demonstrate strong engineering execution. This translates to convincing empirical gains on the ExtremeOcc-3D dataset, where the proposed method achieves noticeably better Point-FID than current leading generators like TRELLIS and SAM3D. If validated more rigorously, this dual-branch consensus mechanism holds genuine promise for real-world perception systems.

**Weaknesses**

Despite these practical merits, several technical and methodological concerns constrain the current impact of the work. The theoretical justification linking attention logit smoothing to low-pass preservation relies on strong, empirically unvalidated assumptions, such as the Lipschitz continuity of velocity with respect to logits. Consequently, this analysis reads more as post-hoc intuition than a rigorous formal guarantee, an issue compounded by the lack of clear separation between heuristic arguments and formal proofs in the text. Furthermore, the conceptual novelty leans heavily on integrating existing practices rather than introducing fundamentally new learning principles. On the empirical side, the evaluation of semantic disambiguation is limited by a very small benchmark of only 21 cases, and the metrics predominantly reflect perceptual similarity rather than direct geometric correctness in occluded regions. Finally, the framework's completeness is compromised when adapting to strong backbones like TRELLIS, where the necessary removal of the visibility-aware mask due to missing pose estimation raises valid questions about consistent cross-architecture compatibility, and key implementation details regarding prior quality sensitivity and failure modes remain insufficiently discussed.

---

> ### Author Rebuttal · Authors · 2026-03-31
>
> We thank the reviewer for recognizing the relevance of ambiguity-aware amodal 3D generation, the novelty/value of our training-free dual-branch framework, the engineering quality, and the gains on challenging benchmarks.
>
> # W1: Assumptions / formal analysis
>
> Agreed. In revision, we will separate the OT guarantees (Theorems A8,A9) from the heuristic/spectral discussion, and better justify the assumptions with prior work and new evidence:
>
> - Our assumptions are mathematically grounded: Lipschitz continuity of Transformer attention (Assumption A.11) is a common basis for regularization (Ye et al. 2023; Geshkovski et al. 2025). Band-limited semantic transport is also consistent with neural spectral bias (Rahaman et al. 2019; Xu et al. 2025).
> - Empirically, logit blur increases attention entropy by 9.1% and reduces logit high-frequency ratio/TV/variance by 77.1%/55.0%/22.8%; the induced prior velocity further reduces TV/norm/local sensitivity by 9.3%/17.4%/2.25% on all 10 tested cases.
>
> These results support that logit smoothing induces the vector-field relaxation in our theory. We will add these statistics and citations.
>
> # W2: Novelty
>
> RelaxFlow addresses text-driven amodal 3D generation with a theoretically motivated training-free framework. Our key insight is that observation preservation and semantic disambiguation require different control granularities, motivating the dual-branch design. Its core mechanism is principled: semantic relaxation corresponds to low-pass filtering of the generative vector field, reducing semantic estimation error and improving stability. Multi-prior consensus and time/visibility-aware fusion reduce proxy mismatch and apply semantic guidance only where/when needed. The training-free design is also valuable because it is plug-and-play across backbones. We will clarify this positioning.
>
> # W3&Q2: AmbiSem-3D scale / statistical significance
>
> Thanks for the suggestion. We report CIs and significance tests for both the user study and quantitative metrics, and extend the ambiguity benchmark with additional curated cases.
>
> ||Ours|Best base|95% CI|
> |-|-|-|-|
> |Align|73.91%|11.41%|[70.37, 77.16]|
> |3D Fid.|63.13%|13.59%|[59.32, 66.77]|
> |Overall|68.52%|9.22%|[65.92, 71.00]|
>
> Across 32 participants and 20 cases with two questions each, our method was preferred in 68.5% of judgments (877/1280; 95% Wilson CI: 65.9%-71.0%). A Friedman test across five methods was significant ($\chi^2(4)=63.29$, $p=5.9\times10^{-13}$). Post-hoc one-sided Wilcoxon tests on per-participant preference rates showed that our method significantly outperformed each baseline after Holm correction (all $p=1.89\times10^{-6}<3.3\times10^{-5}$).
>
> Against SAM3D, using 20,000-bootstrap 95% CIs and one-sided paired Wilcoxon tests, our method is significantly better on all ExtremeOcc-3D metrics:
>
> ||SAM3D|Ours|95% CI|p|
> |-|-|-|-|-|
> |CLIP-img|0.84|0.87|[0.868, 0.879]|6.3e-15|
> |CLIP-txt|24.08|27.26|[26.981, 27.535]|4.3e-42|
> |LPIPS|0.54|0.51|[0.492, 0.516]|1.6e-7|
>
> On AmbiSem-3D, our method significantly improves CLIP-txt, while the CLIP-img gain is positive but not significant, consistent with the benchmark being mainly semantically ambiguous rather than heavily occluded.
>
> ||SAM3D|Ours|95% CI|p|
> |-|-|-|-|-|
> |CLIP-img|0.85|0.87|[0.851, 0.875]|0.107|
> |CLIP-txt|26.29|27.23|[26.597, 27.860]|7.7e-4|
>
> We also add extended 100-case & evaluation; see response to Reviewer zk2k W3. Full results will be added in revision.
>
> # W4&Q1: TRELLIS without visibility-aware masking
>
> The visibility-aware mask is optional, not the sole source of improvement. Even without it, our method improves over the backbone; with pose, the full model improves further. In Fig. 5a, SAM3D vs Ours (w/o vis mask) vs Ours (w/ vis mask) is 100.38 vs 92.3 vs 81.11.
>
> # Q3: Region-wise 3D metrics
>
> We additionally report region-wise 3D metrics on ExtremeOcc-3D, where GT meshes are available. We split the GT surface into front-most and camera-hidden regions using the input-view camera, align with ICP, and transfer GT region labels by nearest-neighbor matching. Our full model improves both visible and hidden geometry over the baseline:
>
> ||Global CD(all verts)↓|Hidden CD(surface)|Front-most CD|Hidden voxel IoU↑|
> |-|-|-|-|-|
> |SAM3D|0.12|0.06|0.09|0.06|
> |Ours|0.07|0.04|0.06|0.10|
> |Relative gain|41.4%|30.0%|34.2%|56.2%|
>
> Thus, the gain is not confined to front-surface fitting, but extends to hidden geometry. This split is object-centric, not fully scene-aware: the “visible” subset is the front-most surface of the isolated object under the input-view camera, and may still be occluded in the scene image. This also explains why visible-region metrics can improve, as better hidden-part completion can improve global shape consistency and surface continuity.
>
> # Limitation
>
> We agree that prior-quality sensitivity and representative failure modes should be stated more explicitly. We add an analysis for low-quality priors (see response to Reviewer zk2k-Q1). We will state these more clearly.

---

> > ### Author Rebuttal · Reviewer_JzMo · 2026-04-03
> >
> > - The additional statistical analysis for AmbiSem-3D is useful and improves confidence in the reported results. However, the benchmark is still relatively small and curated, so I still have some concern about how broadly the conclusions generalize.
> >
> > - The clarification on backbone adaptation is appreciated, but I still think the missing visibility-aware component in TRELLIS leaves some uncertainty about how fully the framework transfers across different architectures.
> >
> > Currently, I decide to maintain my score.

---

> > > ### Author Response · Authors · 2026-04-06
> > >
> > > Thank you again for the positive assessment and constructive follow-up questions. We clarify both points below and narrow our claim where appropriate.
> > >
> > > - Size of AmbiSem-3D
> > >
> > > The original 21-case AmbiSem-3D set is limited in scale if viewed in isolation. Our intent, however, is for AmbiSem-3D to serve as a high-precision diagnostic set for semantic ambiguity under fixed visible evidence. To strengthen this part, we further extend it with 100 additional cases; we briefly restate the results here for completeness. The original 21 cases focus on inherent ambiguity, while the extension also covers view-induced ambiguity. All added cases are semi-automatically mined and manually filtered, and the trend remains consistent:
> > >
> > > Tab A3: Extended 100 cases
> > >
> > > |       | CLIP-img | CLIP-txt |
> > > | ----- | -------: | -------: |
> > > | SAM3D |     83.9 |     25.6 |
> > > | Ours  | **84.5** | **26.9** |
> > >
> > > This diagnostic set also complements the larger ExtremeOcc-3D benchmark (264 cases). **Since the extension is built using a semi-automatic mining pipeline, the current scale is not a hard limit; the benchmark is naturally extendable.** We will include details of this extended evaluation in the revision and release the full set.
> > >
> > > - Transferability of the full components
> > >
> > > The TRELLIS experiment is intended to show transfer of the core mechanism, namely the dual-branch design and relaxation mechanism. We agree that the claim should be framed more narrowly in this part: the dual-branch design and relaxation mechanism transfer across backbones, while the visibility-aware component is an auxiliary, architecture-dependent refinement that requires a pose or visibility signal. Under this scope, the TRELLIS variant still improves over its baseline, supporting cross-backbone transfer of the core mechanism. We will revise the wording accordingly.
> > >
> > > At the same time, the visibility-aware component can in principle be transferred to other backbones as well. It is omitted in TRELLIS only because the required pose signal is not directly available in that setting. However, **with external pose or orientation estimation, the same component can be applied to TRELLIS and other models as well.** In particular, [1] shows that object orientation can be recovered via training-free analysis-by-synthesis, making it possible to apply our full component to TRELLIS-like backbones as well.
> > >
> > > [1] Yichong Lu et al. *Orientation Matters: Making 3D Generative Models Orientation-Aligned.* NeurIPS 2025.

---

### Official Review · Reviewer_h6hS · 2026-03-13

**Soundness:** 4
**Presentation:** 4
**Significance:** 3
**Originality:** 4
**Overall Recommendation:** 5
**Confidence:** 5

**Summary:**

This paper addresses a meaningful and well-known problem in 3D object generation: generating 3D objects from a single image when parts of the object are occluded. To tackle this issue, the paper takes a novel perspective by incorporating information from a textual description provided by the user to complete and refine the regions that are not visible (occluded) in the input image. In this way, the method produces 3D objects from images that not only adhere to the input image but also follow a user-provided text prompt, thereby reducing the ambiguity caused by occluded regions in the input.

Specifically, the paper proposes a dual-branch strategy: an Observation Branch for strict adherence to the input image, and a Semantic-Prior Branch that captures global semantics represented in the textual description provided by the user. During the 3D generation process, a visibility-aware fusion mechanism is used to ensure that the semantic guidance influences only the occluded regions, while the 3D parts corresponding to visible regions in the input image remain unchanged.

The paper also provides theoretical justification for the proposed strategy for extracting structural guidance. The method is evaluated against two strong state-of-the-art baselines (TRELLIS and SAM3D) both quantitatively and qualitatively. A user study is also conducted which further strengthens the support for the results. Ablation studies of key method components are also included and are well executed.

**Compliance With Llm Reviewing Policy:**

Affirmed.

**Final Justification:**

I initially rated the paper positively, and my concerns, including the primary one regarding the reporting of inference computation time, have been adequately addressed. The paper is well-written, tackles a meaningful and fundamental problem in 3D literature, and presents a clear and convincing methodology along with strong results. Also, the comparisons with competitive baselines (e.g., TRELLIS and SAM3D) effectively position the work within the existing literature. I recommend acceptance of this paper.

**Key Questions For Authors:**

- In Equation (11), the authors propose a simple yet effective way to compute a metric that measures how visible a 3D voxel is, and use it to adjust the velocity field interpolation. First of all, what was the motivation of using this approach? There may be better and more accurate alternative ways to represent visibility.
For example, one possible proxy is ambient occlusion (AO) [1], which measures how exposed points or voxels in a scene are to ambient light, typically darkening corners and contact regions where light is obstructed.
A recent example of leveraging ambient occlusion in a related context is presented in [2], where AO is used in the problem of 3D mesh parameterization. In that work, the goal is to guide the placement of cutting seams on a mesh surface toward less visible (more occluded) regions so that the resulting texturing and appearance of the mesh look more seamless.
Given this, it could be interesting to incorporate this classic and long-standing AO-based guidance from computer graphics into the velocity interpolation process. Such a formulation might potentially provide a stronger signal for detecting occluded regions. I would be interested in hearing the authors’ response and opinion on this possibility.

- Is there any guarantee that the images produced by the multi-prior consensus stage will contain information about the missing or occluded regions in the observation image? If, for instance, those prior images do not include such information, then the method would fail, as mentioned in Appendix G. Am I correct? It would be beneficial to discuss in the main paper whether there is any way to guarantee this.


**References:**

[1] Zhukov, Sergey, Andrei Iones, and Grigorij Kronin. "An ambient light illumination model." Eurographics workshop on rendering techniques. Vienna: Springer Vienna, 1998.

[2] Zamani, Amirhossein, Bruno Roy, and Arianna Rampini. "Unsupervised Representation Learning for 3D Mesh Parameterization with Semantic and Visibility Objectives." The Fourteenth International Conference on Learning Representations (ICLR). 2026.

**Limitations:**

- It appears that the formulation of this work is primarily designed for native 3D feedforward models such as TRELLIS or SAM3D, which rely on flow-matching formulations. As a result, the current approach may be limited to this class of models. It would be valuable for the paper to discuss how the proposed method could be extended or adapted to other 3D generation pipelines. For instance, it would be interesting to consider whether similar ideas could be applied to SDS-based optimization approaches, such as DreamFusion [1].


**References:**

[1] Poole, Ben, et al. "DreamFusion: Text-to-3D using 2D Diffusion." The Eleventh International Conference on Learning Representations (ICLR). 2023.

**Strengths And Weaknesses:**

**Strengths:**

- The paper is well written and easy to follow, and it addresses a meaningful problem in the 3D generation literature.

- The proposed method does not require training or fine-tuning. This is beneficial because it does not introduce significant computational overhead and makes the approach practically feasible. The experimental section also supports this claim by stating that all experiments are conducted on a single NVIDIA A40 GPU.

- The proposed methodology is supported by theoretical analysis, which improves the understanding of the method and strengthens the overall contribution of the work.

- The paper includes extensive experiments and comparisons with two major state-of-the-art methods in the 3D generation literature, namely TRELLIS and SAM3D. The fact that the proposed approach outperforms these baselines is a strong indication that the method is well evaluated and well established.

**Weaknesses:**

- Although the qualitative and quantitative results are promising and demonstrate the strong potential and superiority of the proposed approach, the paper lacks reporting of computational cost. Providing this information would help practitioners better understand the compute resources required to apply this method.

---

> ### Author Rebuttal · Authors · 2026-03-31
>
> We sincerely thank the reviewer for the highly positive assessment and for recognizing multiple core strengths of our work, including the meaningful problem formulation, the clear presentation, the practical training-free design, the theoretical grounding, and the strong empirical validation over major baselines through quantitative, qualitative, user-study, and ablation results.
>
> # On computational cost
>
> Thanks for this suggestion. We report the runtime on a 10-sample random subset using a single NVIDIA RTX A5000 GPU.
>
> || Inference time (s/sample) | Peak GPU memory allocated (GB) | Peak GPU memory reserved (GB) |
> |-|-|-|-|
> | SAM3D|15.59 |18.07 |21.85 |
> | Ours|38.40 |18.60 |22.60 |
> | overhead |2.46x |+2.9% |+3.4% |
>
> Ours takes 38.4s/sample on average vs 15.6s/sample for vanilla SAM3D, while increasing peak GPU memory only modestly from 18.07 GB to 18.60 GB allocated.
>
> Note that due to limitations in GPU memory, we optimize for memory rather than runtime. Runtime increases because our method adds extra sequential denoising passes for fusion. While it leaves room for space-for-time optimization, such as batched dual-branch inference and feature caching.
>
> # On the motivation for the visibility-aware mask and the suggestion of AO
>
> Thanks for this suggestion. Our current visibility term is designed to be observation-conditioned: it estimates which voxels are supported by the current input view, so that semantic guidance only affects genuinely occluded regions.
>
> Ambient occlusion is an interesting alternative, but it measures global exposure rather than camera-conditioned observability under the current input image, while our visibility term is designed to decide whether a voxel is directly supported by the observation and thus should be preserved by the observation branch. In Eq.11, we require a current-view soft visibility for branch fusion, not a general exposure measure. Hence we use a z-buffer-based camera-conditioned weighting, which directly protects observed regions and restricts semantic guidance to genuinely occluded ones. AO may still be a useful auxiliary prior, but it is not a direct substitute for the visibility term used here.
>
> We will clarify this motivation in the revision and discuss AO-based variants as a promising extension.
>
> # On whether priors must contain the missing-region information
>
> Thanks for this question. Our priors are not intended to provide exact missing-view details.
>
> Their role is to provide high-level semantic / structural intent that disambiguates the completion mode; the backbone still performs the actual amodal completion.
>
> We will clarify this point in the paper and discuss the corresponding failure modes more explicitly.
>
> # On extension beyond flow-based feedforward models
>
> Thanks for this suggestion. This is an interesting direction.
>
> Our current implementation targets flow-based feedforward generators, but the core principle—decoupling hard observation constraints from relaxed semantic guidance—may also inspire optimization-based 3D generation frameworks, especially in SDS, where there are similar diffusion compositions. We will discuss this as future direction.

---

> > ### Author Rebuttal · Reviewer_h6hS · 2026-04-01
> >
> > I appreciate the authors’ responses and will maintain my original score. However, since the inference time is nearly twice that of SAM3D, I strongly suggest including these results (both inference time and GPU memory usage comparisons) in the main paper, along with a discussion of the points raised here to provide clearer insight into the computational resources required for applying this method. Additionally, the same comparison with TRELLIS is currently missing and I would recommend including it as well.

---

> > > ### Author Response · Authors · 2026-04-03
> > >
> > > Thank you again for the supportive assessment and for the constructive suggestion. We will include the computational-cost tables and a clearer discussion of runtime/memory trade-offs in the revised manuscript. We also include the corresponding comparison on the TRELLIS backbone, as recommended:
> > >
> > > || Inference time (s/sample) | Peak GPU memory allocated (GB) | Peak GPU memory reserved (GB) |
> > > | -------- | ------------------------: | ---------------: | -----------------: |
> > > | TRELLIS  |                     12.81 |             5.57 |               5.60 |
> > > | Ours     |                     23.31 |             6.02 |               6.07 |
> > > | Overhead |                     1.82× |            +8.0% |              +8.4% |
> > >
> > > On a single NVIDIA RTX A5000 GPU, the paired comparison on 10 random cases gives 23.31 s/sample for our method versus 12.81 s/sample for vanilla TRELLIS, corresponding to a 1.82× runtime overhead, while peak GPU memory increases only modestly.

---

### Official Review · Reviewer_nwq4 · 2026-03-14

**Soundness:** 3
**Presentation:** 2
**Significance:** 2
**Originality:** 3
**Overall Recommendation:** 4
**Confidence:** 3

**Summary:**

This paper introduces the task of "text-driven amodal 3D generation," which addresses the inherent semantic ambiguity in image-to-3D generation caused by severe object occlusion. Existing feedforward models typically struggle with this ambiguity, often collapsing into an "observation-overfitted" mode where they hallucinate implausible unseen geometries based solely on visible pixels. To resolve this, the authors propose RelaxFlow, a training-free dual-branch inference framework that allows users to guide the completion of occluded regions using text prompts while preserving the visible input observation.The core insight of the work is that preserving the input observation and following a text prompt require distinct control granularities: the observation demands rigid, pixel-level control, while the text prompt requires a relaxed, structural control to accommodate the visible constraints.

**Compliance With Llm Reviewing Policy:**

Affirmed.

**Final Justification:**

Thank you for conducting the additional experiments to address my concerns. I have updated my scores to weak accept. If paper is accepted, please ensure that you incorporate the necessary revisions, such as a discussion of the observed visual artifacts, a clearer explanation of the differences compared to AModal3R, and new 2D editing baselines.

**Key Questions For Authors:**

See Strengths and Weaknesses for details. I like the methodology (see Strengths), but the motivation, introduction, experimental details, and empirical results are not strong (see Weaknesses). I may be willing to raise my score depending on the rebuttal.

**Limitations:**

yes

**Strengths And Weaknesses:**

### Strength
* **Insightful Formulation:** The core concept of applying low-pass filtering to the generative vector field is quite elegant. Selectively suppressing high-frequency instance details to isolate the geometric structure that accommodates the observation is a clever and theoretically grounded approach to the amodal completion problem.

### Weakness
* **W1: Questionable Core Premise (Fundamental Limitation vs. Training Artifact):** The paper claims that feedforward image-to-3D models inherently collapse to an "observation-overfitted" generation. However, this may simply be a limitation of the specific models chosen (which are typically trained exclusively on text-to-3D or image-to-3D), rather than a theoretical limitation of feedforward architectures. Natively multimodal models (e.g., Tripo3D) that are trained simultaneously on image and text conditions do not suffer from this to the same degree. This raises the concern that the proposed training-free framework might just be a "band-aid" for current, limited public models rather than a necessary solution for state-of-the-art or upcoming multimodal generators. Even for the current model that trained jointly with image-to-3D or text-to-3D but not image+text-to-3D, it is still worthy showing their results as baselines, also see W4 for an example of asked ablation study.

* **W2: Unclear and Potentially Weak 2D Baselines:** The details regarding the 2D baseline are insufficient. It is not clear which specific 2D editing pipeline was used to generate the baseline results in Fig.4. It is hard to find information regarding to it. Furthermore, when independently testing the input images and text prompts from Figure 4  using modern state-of-the-art 2D generation/editing models (e.g., Nano Banana 2), the reviewer found the 2D outputs are significantly better than what is reported in the paper, suggesting the baselines may not represent a fair comparison. Basically, modern image editing model can handle all input observations in Fig.4 and generate new images that follow both observation and intent text prompt.

* **W3: Misaligned Motivation and Positioning:** The introduction heavily focuses on the limitations of standard text-to-3D and image-to-3D models for occluded views. This framing would be appropriate if this were the very first work in amodal 3D reconstruction. However, given the existence of occlusion-aware models like Amodal3R, the motivation feels slightly disconnected. The paper should explicitly compare to Amodal3R, describe its limitation, and show your theoretical improvement/solution. A better flow might be: why amodal 3D is important --> what is the state-of-the-art amodal 3d model and their limitation? --> what findings/insights/theories motivate your contributions? --> what is your contribution? --> what are the results.  Now, after reading the current paper, the reviewer still has concern why not improve upon the existing feedforward model Amodal3R for this problem? In 2D, occlusion/ambiguity is seamlessly handled via inpainting models, which intentionally lock the observation and use text intent to fill missing regions. Amodal3R essentially attempts this in 3D. The authors need to better justify why a complex dual-branch, optimization/inference-time intervention is superior to a native 3D inpainting approach.

* **W4: Missing Crucial Ablation for the Semantic Branch:** The paper uses a "Multi-Prior Consensus" module to convert text into image proxies for the semantic branch. If TRELLIS already natively supports text conditioning and image conditioning (but not at the same time), the authors should still include a baseline ablation: a two-branch setup where the second (semantic) branch uses the *text prompt directly*, rather than relying on the consensus of multiple queried/generated images. A different CFG or weighting can be applied.


* **W5: Visual Artifacts and Inaccuracies:** There are noticeable geometric and color inconsistencies in the generated outputs that are not addressed in the text. For example, the proportion of the generated sofa in Figure 1 is not aligned with input. In Figure 4, the top example fails to match the expected backpack color, and the second example exhibits artifacts on the bicycle's handlebar.


### Minor Comments
* **Separable Convolution Filter:** Could the authors briefly clarify the rationale for using a *separable* convolution filter? Is it purely for computational efficiency, or does it uniquely benefit the attention matrix dimensions?
* **Missing Benchmark Examples:** The paper introduces two new diagnostic benchmarks, ExtremeOcc-3D and AmbiSem-3D, but comprehensive visual examples or a full gallery of the benchmark cases are missing. Releasing more examples (or the dataset itself during the review period) would greatly help in assessing the diversity and difficulty of the curated ambiguities.

---

> ### Author Rebuttal · Authors · 2026-03-31
>
> We thank the reviewer for the detailed feedback and for recognizing the theoretical grounding of our low-pass formulation. To address the concerns, we add stronger comparisons to Nano Banana Pro and TRELLIS text+image variant, and clarify key points, such as the distinct task, the benchmark examples and baseline details.
>
> # W1: Core Premise (Fundamental Limitation vs. Training Artifact)
> Thanks for this point. Our claim is not an architecture-level impossibility for all feedforward generators, but that such models are not obviously immune: under severe occlusion or ambiguity, reliable disambiguation still depends on broad image-text-3D training coverage, which becomes harder as modalities and condition combinations grow. More importantly, text-driven amodal 3D generation is a distinct and meaningful task: given fixed visible evidence, text specifies the intended hidden semantics. Thus, our contribution is not only a remedy for current models, but the formulation and solution of this task via a practical, theoretically motivated training-free framework. We will revise the manuscript to make this clearer.
>
> # W2: 2D editing baseline
> Thanks for pointing this out. As stated in the experiments, the 2D editing baseline uses SDXL, which performed best among the open-source 2D editing models we tested. We will clarify this in the main text and captions. More importantly, our target is not only single-view editing quality, but observation-consistent 3D completion across views. Even strong 2D editors are not 3D-aware, so they do not enforce cross-view consistency or strict preservation of the observed evidence, which is a key strength of our method.
>
> We also report a direct comparison with Nano Banana Pro in Tabs. A1 and A2 (in response to Reviewer zk2k), and qualitatively at [[link]](https://anonymous.4open.science/r/5749/comp-nbp.png). The qualitative results show the drawback of non-3D-aware 2D editing, even for this SOTA commercial model. Quantitatively, we achieve better CLIP-img with competitive CLIP-txt.
>
> # W3: Motivation relative to Amodal3R
> Thank you for this suggestion. We believe the main confusion is that Amodal3R and our method target different goals. Amodal3R learns a fixed amodal completion prior from training data, i.e., it predicts the most likely hidden geometry under that prior. Our setting instead focuses on semantic branching under fixed visible evidence: the same observation may admit multiple plausible amodal completions, and the user specifies the desired one via text. Thus, the key challenge is controllable disambiguation, not only completion.
>
> This is why a native amodal feedforward model is insufficient for our setting. It can complete occluded regions, but does not let the user control which plausible completion is produced. Our method is designed for this missing capability: it preserves observed evidence with an observation branch, while injecting text-derived intent through a separate semantic branch with relaxed guidance.
>
> We also directly compare against Amodal3R in Tab. 1 in the main paper and outperform it on ExtremeOcc-3D (e.g., P-FID 81.11 vs 129.46; CLIP-txt 27.26 vs 22.29). Since RelaxFlow is training-free and plug-and-play, it could in principle also be combined with an amodal feedforward backbone such as Amodal3R, which we will clarify in the revision.
>
> # W4: Semantic-branch ablation
> We add a TRELLIS baseline in which the semantic branch uses direct text conditioning with CFG-style interpolation. As shown in Tabs. A1 and A2 (in response to Reviewer zk2k), our method consistently outperforms this TRELLIS-text+image baseline. We also provide qualitative comparisons at [[link]](https://anonymous.4open.science/r/5749/comp-Trellis.png). This is expected because TRELLIS’s text- and image-conditioned variants are trained separately, so naive composition leads to semantic misalignment.
>
> # W5: Visual artifacts
> Thanks for pointing this out. Color shift and detail mismatch are inherent issues of existing feedforward 3D generators. In our design, we consistently improve semantic alignment while better preserving visible evidence than competing baselines. We will add a more explicit failure-case discussion.
>
> # Q1: Separable Gaussian filter
> Thanks for this question. The separable implementation is mainly for efficiency and scalability on large attention maps, while realizing the intended low-pass smoothing over attention logits.
>
> # Q2: Benchmark examples
> Thanks for the suggestion. We provide all benchmark cases at [[link]](https://anonymous.4open.science/r/5749/cases.png) and will release the full benchmark resources publicly.
>
> Overall, we will revise the positioning to better distinguish our task from native amodal completion, clarify baseline details in the main text, and strengthen the empirical section with the requested ablations and stronger comparisons. We hope these additions address the reviewer's concerns and would be grateful if the reviewer could reconsider the score.

---

> > ### Author Rebuttal · Reviewer_nwq4 · 2026-04-06
> >
> > The authors conducted the additional experiments that have fully addressed my concerns.

---

> > > ### Author Response · Authors · 2026-04-06
> > >
> > > Thank you very much for the positive update. We are glad that the additional experiments and clarifications addressed your concerns, and we sincerely appreciate your reconsideration.

---

### Official Review · Reviewer_zk2k · 2026-03-17

**Soundness:** 2
**Presentation:** 3
**Significance:** 3
**Originality:** 3
**Overall Recommendation:** 4
**Confidence:** 4

**Summary:**

This paper focuses on text-driven amodal 3D generation, where text prompts are used to resolve ambiguity in heavily occluded image-to-3D generation while preserving the visible evidence from the input image. To address this, the paper proposes RelaxFlow, a training-free dual-branch inference framework. One branch enforces observation fidelity, while the other branch injects semantic guidance through text-derived visual priors. The method further introduces multi-prior consensus and a low-pass relaxation mechanism implemented by smoothing cross-attention logits in the semantic-prior branch. The paper also presents two new benchmarks, ExtremeOcc-3D and AmbiSem-3D, and reports improvements over SAM3D- and TRELLIS-based baselines.

**Compliance With Llm Reviewing Policy:**

Affirmed.

**Final Justification:**

This paper combines several techniques to improve text-driven amodal 3D generation. The additional comparisons with stronger baselines provided in the rebuttal are convincing. Therefore, I raise my score to weak accept.

**Key Questions For Authors:**

1. How sensitive is the method to low-quality or mismatched prior images, especially in the generated-prior setting?

2. I am curious about a stronger baseline: using SOTA video models to provide complete multi-view images and then reconstruct or run multi-view to 3D models

I would like to raise my score if the authors could address my concerns.

**Strengths And Weaknesses:**

Strengths:

1. In heavily occluded image-to-3D generation, the visible evidence is often insufficient to determine a unique amodal completion, and allowing the user to resolve this ambiguity through text is a natural and meaningful direction.

2. The motivation is reasonable. The key design choice—decoupling strict observation preservation from relaxed semantic guidance—makes intuitive sense, and the dual-branch formulation is consistent with the paper’s motivating argument about different control granularities.

3. The paper introduces two diagnostic benchmarks: ExtremeOcc-3D and AmbiSem-3D, reports quantitative results, includes a user study on the ambiguous-setting benchmark, and provides ablation studies on the main design choices.

Weaknesses:

1. The method mainly combines several intuitive inference-time ingredients: text-to-visual prior conversion, multi-prior consensus, cross-attention smoothing, and time/visibility-aware branch fusion. While the combination is sensible and effective, it feels more like a carefully designed system pipeline than a fundamentally new modeling contribution.

2. The analysis relies on strong assumptions, such as band-limited semantic transport and high-frequency-dominated estimation error, and the practical implementation operates by smoothing attention logits rather than directly filtering the vector field. As a result, the theory feels more like an intuition-supporting abstraction than a rigorous explanation of why the method works in the actual backbone models.

3. The new benchmarks are useful, but still relatively limited in scale. AmbiSem-3D contains only 21 ambiguous cases. Given that semantic branching is one of the central claims of the paper, this benchmark feels too small to fully establish generality.

4. This paper needs a stronger baseline: using SOTA video/multi-view models to provide complete multi-view images and then reconstruct 3D objects, or directly run multi-view to 3D models.

---

> ### Author Rebuttal · Authors · 2026-03-31
>
> We thank the reviewer for recognizing the motivation, dual-branch intuition, and benchmark/ablation value of our work. To address the concerns, we add stronger video/multiview baselines, prior-quality sensitivity analysis, an extended 100-case ambiguity evaluation, and a clearer logit-smoothing/velocity-relaxation explanation.
>
> # W1: Design & contribution
>
> RelaxFlow addresses text-driven amodal 3D generation with a theoretically motivated training-free framework. Our key insight is that observation preservation and semantic disambiguation require different control granularities, motivating the dual-branch design. Its core mechanism is principled: semantic relaxation corresponds to low-pass filtering of the generative vector field, improving stability and reducing semantic estimation error. Multi-prior consensus and time/visibility-aware fusion implement this by reducing proxy mismatch and applying semantic guidance only where needed. The training-free design is also valuable, making RelaxFlow plug-and-play across backbones. We will clarify this positioning.
>
> # W2: Theory
>
> We agree that logit smoothing is a practical approximation to vector-field filtering, but the link is mathematically grounded.
>
> * In Appen. A4, under Assumption A.11, logit smoothing induces a controlled relaxation of the resulting vector field $\tilde{v}_{\theta}$, consistent with prior analyses of Transformer regularity/stability [1,2].
> * Empirically, logit blur increases attention entropy by 9.1% and reduces logit high-frequency ratio/TV/variance by 77.1%/55.0%/22.8%; the induced prior velocity further reduces TV/norm/local sensitivity by 9.3%/17.4%/2.25% on all 10 tested cases.
> * More broadly, our band-limited semantic transport premise is supported by neural spectral bias [3,4], favoring low-frequency global structure over high-frequency instance noise.
>
> We will add this clarification and evidence.
>
> [1] Ye et al., UAI 2023.
> [2] Geshkovski et al., Bull. AMS 2025.
> [3] Rahaman et al., ICML 2019.
> [4] Xu et al., CAMC 2025.
>
> # W3: Scale of AmbiSem-3D
>
> AmbiSem-3D is a high-precision diagnostic set for semantic ambiguity under fixed visible evidence, not a large standalone benchmark. To address the concern that the original 21-case set is small, we extend 100 additional cases. The original set focuses on inherent ambiguity; the extended set also includes view-induced ambiguity. All added cases are semi-automatically mined and manually filtered. The trend remains consistent:
>
> Tab A3: Extended 100 cases
>
> ||CLIP-img|CLIP-txt|
> |-|-|-|
> |SAM3D|83.9|25.6|
> |Ours|**84.5**|**26.9**|
>
> This complements the larger ExtremeOcc-3D benchmark (264 cases). We will include the extension and release the benchmark.
>
> # W4&Q2: Video/multiview baselines
>
> We additionally evaluate two SOTA baselines, Wan2.2-TI2V [5] and MV-Adapter [6]. We use the input image+text to generate multi-view evidence, then reconstruct 3D with TRELLIS's multi-image pipeline. As shown in Tabs. A1 and A2, ours outperforms both baselines consistently. Qualitative comparisons are also provided at [[**link**]](https://anonymous.4open.science/r/5749/comp-wan.png). These video/multiview models still show weaker 3D consistency, while our method is natively 3D.
>
> Tab A1: ExtremeOcc-3D
>
> ||CLIP-img|CLIP-txt|FID↓|LPIPS↓|P-FID↓|
> |-|-|-|-|-|-|
> |Ours|**0.87**|27.26|**39.44**|**0.51**|**81.1**|
> |Wan-TI2V+3D|0.78|22.98|107.49|0.85|110.3|
> |MV-Adapter+3D|0.80|22.70|148.66|0.81|238.3|
> |Trellis-t+i|0.82|26.36|116.51|0.79|137.7|
> |Nano Banana Pro+3D|0.83|**27.49**|77.68|0.77|88.6|
>
> Tab A2: AmbiSem-3D
>
> ||CLIP-img|CLIP-txt|
> |-|-|-|
> |Ours|**0.87**|27.23|
> |Wan-TI2V+3D|0.80|26.71|
> |MV-Adapter+3D|0.81|25.54|
> |Trellis-t+i|0.81|27.24|
> |Nano Banana Pro+3D|0.81|**27.41**|
>
> [5] Ang et al., 2025.
> [6] Huang et al., ICCV 2025.
>
> # Q1: Sensitivity to low-quality/mismatched priors
>
> Replacing the priors on AmbiSem-3D with a much weaker T2I model, SD1.5 instead of Z-Image, reduces CLIP-img/CLIP-txt by only 0.02/0.7, indicating moderate degradation.
>
> For mismatched priors, our method assumes broad compatibility between observation and prior: the prior should resolve ambiguity, not contradict visible evidence. Under strong conflict, there may be no consistent 3D solution, and the model can behave more like soft concept fusion (Appen. Fig. 11). This is also why we use multiple priors: consensus suppresses isolated poor priors and reduces the chance that one mismatched prior dominates. We will clarify both the robustness and this limitation.
>
> We hope these additions address the reviewer's concerns and would be grateful if the reviewer could reconsider the score.

---

> > ### Author Rebuttal · Reviewer_zk2k · 2026-04-03
> >
> > I appreciate your rebuttal. My concerns have been addressed. The added comparisons with stronger baselines are convincing, so I will raise my score.

---

> > > ### Author Response · Authors · 2026-04-06
> > >
> > > Thank you very much for the encouraging follow-up and for raising your score. We are glad that the added experiments and stronger baseline comparisons helped address your concerns.

---

### Decision · Program_Chairs · 2026-04-30

**Decision:**

Accept (spotlight)

**Comment:**

This paper was reviewed by 4 experts in the field. After discussion, the reviewers still hold a consistent review to this work. The rating is 4(weak accept), 4(weak accept), 4(weak accept), 5(accept).

In general, reviewers agree that this a high-quality work. Particularly, it 1) provides a practical training-free framework, 2) introduces a novel dual-branch design using spectral relaxation, and 3) delivers strong empirical results on new specialized benchmarks. Area chair also agrees that this is one of few work with simple and clean theory and shows effectiveness in practice.

Reviewers raised several concerns to this work. The concerns include 1) the reliance of the theoretical justification on unvalidated assumptions like Lipschitz continuity, and 2) issues with cross-architecture compatibility and the core premise of observation-overfitting. Still, none of these are critical.

Based on this, the decision of this work is to Strong accept. Still, we strongly recommend the authors carefully read all reviewers’ final feedback and revise the manuscript as suggested in the final camera-ready version if being accept.